



# IntelliO3-ts v1.0: A neural network approach to predict near-surface ozone concentrations in Germany

Felix Kleinert[1,2], Lukas H. Leufen[1,2], and Martin G. Schultz[1]

[1]Forschungszentrum Jülich, Jülich Supercomputing Center, Wilhelm-Johnen-Straße 52428 Jülich
[2]Rheinische Friedrich-Wilhelms-Universität Bonn, Institute of Geosciences, Germany

**Correspondence:** Felix Kleinert (f.kleinert@fz-juelich.de)





**Abstract.**

The prediction of near-surface ozone concentrations is important to support regulatory procedures for the protection of humans from high exposure to air pollution. In this study, we introduce a data-driven forecasting model named 'IntelliO3-ts', which consists of multiple convolutional neural layers (CNN), grouped together as inception blocks. The model is trained

with measured multi-year ozone and nitrogen oxides concentrations of more than 300 German measurement stations in rural environments, and six meteorological variables from the meteorological COSMO reanalysis. This is by far the most extensive dataset used for time series predictions based on neural networks so far. IntelliO3-ts allows predicting daily maximum 8-hour average (dma8eu) ozone concentrations for a lead time of up to four days, and we show that the model outperforms standard reference models like persistence. Moreover, we demonstrate that IntelliO3-ts outperforms climatological reference models for

the first two days, while it does not add any genuine value for longer lead times. We attribute this to the limited deterministic information that is contained in the single station time series training data. We applied a bootstrapping technique to analyse the influence of different input variables and found, that the previous day ozone concentrations are of major importance, followed by 2m temperature. As we did not use any geographic information to train IntelliO3-ts in its current version and included no relation between stations, the influence of the horizontal wind components on the model performance is minimal. We expect

that the inclusion of advection-diffusion terms in the model could improve results in future versions of our model.

# 1  Introduction

Exposure to ambient air pollutants such as ozone ($O_3$) is harmful for living beings (WHO, 2013; Bell et al., 2014; Lefohn et al., 2017; Fleming et al., 2018) and certain crops (Avnery et al., 2011; Mills et al., 2018). Therefore, the prediction of ozone concentrations is of major importance to issue warnings for the public if high ozone concentrations are foreseeable. As tropo-

20 spheric ozone is a secondary air pollutant, there is nearly no source of directly emitted ozone. Instead, it is formed in chemical reactions of several precursors like nitrogen oxides ($NO_x$) or volatile organic compounds (VOCs). Weather conditions (temperature, irradiation, humidity, and winds) have a major influence on the rates of ozone formation and destruction. Ozone has a "chemical lifetime" in the lower atmosphere of several days and can, therefore, be transported over distances of several hundred kilometres.

Ozone concentrations can be forecasted by various numerical methods. Chemical transport models (CTMs) solve chemical and physical equations explicitly (for example Collins et al., 1997; Wang et al., 1998a, b; Horowitz et al., 2003; von Kuhlmann et al., 2003; Grell et al., 2005; Donner et al., 2011). These numerical models predict concentrations for grid cells, which are assumed to be representative for a given area. Therefore, local fluctuations which are below model resolution cannot be simulated. Moreover, CTMs often have a bias in concentrations, turnover rates or meteorological properties which have a direct

influence on chemical processes (Vautard, 2012; Brunner et al., 2015).

This makes CTMs unsuited for regulatory purposes, which by law are bound to station measurements, except if so-called model output statistics are applied to the numerical modelling results (Fuentes and Raftery, 2005). As an alternative to CTMs, regression models are often used to generate point forecasts (c.f. Olszyna et al., 1997; Thompson et al., 2001; Abdul-Wahab





et al., 2005). Regression models are pure statistical models, which are based on empirical relations among different variables.
They usually describe a linear functional relationship between various factors (precursor concentrations, meteorological, and
site information) and the air pollutant in question.

Since the late nineties machine learning techniques in the form of neural networks have also been applied as a regression
technique to forecast ozone concentrations or threshold value exceedances (see Table 1). As the computational power was
limited in the early days of those approaches, many of these early studies focused on a small number of measurement stations
and used a fully connected (FC) network architecture. More recent studies explored the use of more advanced network archi-
tectures like convolutional neural networks (CNN) or Long-Short-Term Memory networks (LSTM). These networks were also
applied to a larger number of stations compared to the earlier studies and some studies have tried to generalise, i.e. to train
one neural network for all stations instead of training individual networks for individual stations (Table 1). Although the total
amount of studies focusing on air quality or explicit near-surface ozone is already quite substantial, there are only few studies
which applied advanced deep learning approaches on a larger number of stations or on longer time series. Eslami et al. (2019)
applied a CNN on time series of 25 measurement stations in Seoul, South Korea to predict hourly ozone concentrations for the
next 24 hours. Ma et al. (2020) trained a bidirectional LSTM on 19 measurement sites over a period of roughly 9 months, and
afterwards used that model to retrain individually for 48 previously not used measurement stations (transfer learning).

Sayeed et al. (2020) applied a deep CNN on data from 21 different measurement stations over a period of four years.
They used three years (2014 to 2016) to train their model and evaluated the generalisation capability on the fourth year (2017).
Zhang et al. (2020) developed a hybrid CNN-LSTM model to predict gridded air quality concentrations ($O_3$, $NO_2$, $CO$, $PM_{2.5}$,
$PM_{10}$).

The current study extends these previous works and introduces a new deep learning model for the prediction of daily maxi-
mum 8-hour average $O_3$ concentrations (dma8eu, see Sect. 2.1) for a lead time of up to four days. The network architecture is
based on several stacks of convolutional neural networks. We trained our network with data from 313 background measurement
stations in Germany (date range from 1997 to 2007), tuned hyperparameters on data from 212 stations (data rage from 2008 to
2009), and finally we evaluate the performance on 204 stations, which have data during the 2010 to 2015 period.

This article is structured as follows: In Sect. 2 we explain our variable selection and present our prepossessing steps. In Sect.
3, we introduce our forecasting model IntelliO3-ts, version 1.0. Sect. 4 introduces the statistical tools and reference models,
which were used for verification. In Sect. 5 we present and discuss the results and analyse the influence of different input
variables ob the model performance. Finally, Sect. 6 provides conclusions.

## 2 Variable selection and data processing

### 2.1 Variable selection

Tropospheric ozone ($O_3$) is a greenhouse gas formed in the atmosphere by chemical reactions of other directly emitted pollu-
tants (ozone precursors) and therefore referred to as a secondary air pollutant.

The continuity equation of near surface ozone in a specific volume of air can be written as (Jacobson, 2005, p.74f):





$$\frac{\partial N_q}{\partial t} + \nabla \cdot (\mathbf{v} N_q) = (\nabla \cdot \mathbf{K_h} \nabla) N_q + R_{\text{depg}} + R_{\text{chemg}}, \tag{1}$$

where $\frac{\partial N_q}{\partial t}$ is the partial derivative of the ozone number concentration with respect to time, $\mathbf{v}$ is the vector wind velocity, $\mathbf{K_h}$ is the eddy diffusion tensor for energy, while $R_{\text{depg}}$ and $R_{\text{chemg}}$ are the rates of dry deposition to the ground, and photo-chemical production or loss, respectively.

Tropospheric ozone is formed under sunlit conditions in gas-phase chemical reactions of peroxy radicals and nitrogen oxides (Seinfeld and Pandis, 2016). The peroxy radicals are themselves oxidation products of volatile organic compounds. The nitrogen oxides undergo a rapid catalytic cycle:

$$NO_2 + h\nu + O_2 + M \rightarrow NO + O_3 + M^* \tag{R1}$$

$$NO + O_3 \rightarrow NO_2 + O_2, \tag{R2}$$

where NO and $O_3$ are converted to $NO_2$ and back within minutes (M is an arbitrary molecule which is needed to take up excess energy, denoted by the asterisk). As a consequence, ozone concentrations in urban areas with high levels of $NO_x$ from combustive emissions are extremely variable. In this study, we therefore focus on background stations, which are less affected by the rapid chemical inter-conversion.

From a chemical perspective, the prediction of ozone concentrations would require concentration data of NO, $NO_2$, speciated VOC, and $O_3$ itself. However, since speciated VOC measurements are only available from very few measurement sites, the chemical input variables of our model are limited to NO, $NO_2$, and $O_3$.

Besides the trace gas concentrations, ozone levels also depend on meteorological variables. Due to the scarcity of reported meteorological measurements at the air quality monitoring sites, we extracted time series of meteorological variables from the 85    6 km resolution COSMO-Reanalysis (Bollmeyer et al., 2015, COSMO-REA6) and treat those as observations.

All data used in this study were retrieved from the Tropospheric Ozone Assessment Report (TOAR) database (Schultz et al., 2017) via the Representational State Transfer (REST) Application Programming Interface (API) at https://join.fz-juelich.de. The air quality measurements were provided by the German Umweltbundesamt, while the meteorological data were extracted from the COSMO-REA6 reanalysis as described above. These reanalysis data cover the period from 1995 to 2015 with some 90    gaps due to incompleteness in the TOAR database. As discussed in the US EPA guidelines on air quality forecasting (Dye, 2003), ozone concentrations typically depend on temperature, irradiation, humidity, wind speed and wind direction. The vertical structure of the lowest portion of the atmosphere (i.e. the planetary boundary layer) also plays an important role, because it determines the rate of mixing between fresh pollution and background air. Since irradiation data were not available from the REST interface, we used cloud-cover together with temperature as proxy variables.

Table 2 shows the list of input variables used in this study, and Table 3 describes the daily statistics that were applied to the hourly data of each variable.





As described above, ozone concentrations are less variable at stations, which are further away from primary pollutant emission sources. We therefore selected those stations from the German air quality monitoring network, which are labelled as "background" stations according to the European Environmental Agency (EEA), Airbase classification.

## 2.2 Data processing

We split the individual station time series into three non-overlapping time periods for training, validation and testing which we will refer by *set* from now on (see Fig. 2). We only used stations which at least have one year of valid data in one set. Firstly, the time span of the training data set is ranging from Jan 1st, 1997 to Dec 31st, 2007. Secondly, the validation set is ranging from Jan 1st, 2008 to Dec 31st, 2009. Thirdly, the test set ranges from Jan 1st, 2010 to Dec 31st, 2015.

Due to changes in the measurement network over time, the number of stations in the three datasets differ: training data comprises 313 stations, validation data 212 stations, and testing 204 stations. This is by far the largest air quality time series dataset that has been used in a machine learning study so far (see Table 1).

Supervised learning techniques require input data ($\mathbf{X}$) and a corresponding label ($\boldsymbol{y}$) which we create for each station of the three sets as depicted in Algorithm 1.

---

**Algorithm 1** Data preprocessing

---

1: Standardise time series to approximately mean zero and unit variance (approximation for z-transformation)

2: Linearly interpolate missing data (maximally one missing data point between valid samples)

3: **for all** Stations: Create samples (from standardised time series) **do**

4:    Create inputs $\mathbf{X}$ with variables of seven days ($-6d$) to ($0d$).

     Shape of $\mathbf{X}$: 7 by 1 by 9 (number of days, 1, number of variables)

5:    Create labels $\boldsymbol{y}$ with ozone (dma8eu) concentrations for the next four days (1d to 4d).

     Shape of $\boldsymbol{y}$: 1 by 4 (1, lead time)

6: **end for**

7: Remove all $\mathbf{X}, \boldsymbol{y}$ pairs which include any missing value

8: In the train set duplicate extremes (samples, where $y_i < -3$ or $y_i > 3$)

9: Permute samples in the train set (and only in the train set)

10: Create batches (collection of samples) of size 512

---

Samples within the same data set (train, validation, test) can overlap which means that one missing data point would appear up to seven times in the inputs $\mathbf{X}$ and up to four times in the labels $\boldsymbol{y}$. Consequently, one missing value will cause the removal of eleven samples (Algorithm 1, line 7). As we want to keep the number of samples as high as possible, we decided to linearly interpolate (Algorithm 1, line 2) the time series if only one consecutive value is missing. Table 4 shows the number of stations per data set (train, val, test) and the corresponding amount of samples (pairs of inputs $\mathbf{X}$ and labels $\boldsymbol{y}$) per data set. Moreover, Table A1 summarises all samples per station individually. Figure 1 shows a map of all station locations.





We trained the neural network (details on the network architecture are given in Sect. 3) with data of the train set and tuned hyperparameters exclusively on the validation data set. For the final analysis and model evaluation, we use the independent test data set, which was neither used for training the models parameters, nor for tuning the hyperparameetrs. Random sampling, as is often done in other machine learning applications, and occasionally even in other air quality or weather applications of machine
learning, would lead to overly optimistic results due to the multi-day auto-correlation of air quality and meteorological time series. Horowitz and Barakat (1979) already pointed to this issue when dealing with statistical tests. Likewise, the alternative split of the dataset into spatially segregated data could lead to the undesired effect that two or more neighbouring stations with high correlation between several paired variables fall into different data sets. Again, this would result in overly optimistic model results.

By applying a temporal split, we ensure that the training data do not directly influence the validation and test data sets. Therefore, the final results reflect the true generalisation capability of our forecasting model.

In accordance with other studies, our initial deep learning experiments with a subset of this data have shown that neural networks, just as other classical regression techniques, have a tendency to focus on the mean of the distribution and perform poorly on the extremes. However, especially the high concentration events are crucial in the air quality context due to their
strong impact on human health and the adverse crop effects. Extreme values occur relatively seldomly in the dataset, and it is therefore difficult for the model to learn their associated patterns correctly. To increase the total number of values on the tails of the distribution during training, we append all samples where the standardised label (i.e. the normalised ozone concentration) is $< -3$ or $> 3$ for a second time on the training data set (Algorithm 1, line 8).

We selected a batch size of 512 samples (Algorithm 1, line 10). Before creating the different training batches, we permute
the ordering of samples per station in the training set, to ensure that the distribution of each batch is similar to those of the full training data set (Algorithm 1, line 9). Otherwise, each batch would have an underrepresented season and consequently would lead to undesired looping during training (e.g. no winter values in the first batch, no autumn values in the second batch, etc.).

## 3 Model setup

Our machine learning model is based on a convolutional layer neural network (LeCun et al., 1998), which was initially designed
for pattern recognition in computer vision applications. It has been shown that such model architectures work equally well on time series data as other neural network architectures, which have been especially designed for this purpose, such as recurrent neural networks or LSTMs (Dauphin et al., 2017; Bai et al., 2018). Schmidhuber (2015) provides a historical review on different deep learning methods, while Ismail Fawaz et al. (2019) focus especially on deep neural networks for time series.

Our neural network named IntelliO3-ts, version 1.0, primarily consists of two inception blocks (Szegedy et al., 2015), which
combine multiple convolutions, execute them in parallel, and concatenate all outputs in the last layer of each block. Figure 3 depicts one inception block, and Figure A2 shows the entire model architecture including the first input layers and final flat and output layers. We treat each input variable (see Table 2) as an individual channel (like R, G, B in images) and use time as the first dimension (this would correspond to the width axis of an image). Inputs ($\mathbf{X}$) consist of the variable values from





7 days (-6d to 0d). Outputs are ozone concentration forecasts (dma8eu) for lead times up to 4 days (1d to 4d). Therefore, we
change the kernel sizes in the inception blocks from $1 \times 1$, $3 \times 3$, and $5 \times 5$, as originally proposed by Szegedy et al. (2015), to
$1 \times 1$, $3 \times 1$, and $5 \times 1$). This allows the network to focus on different temporal relations. The $1 \times 1$ convolutions are also used
for information compression (reduction of the number of filters), before larger convolutional kernels are applied (see Szegedy
et al. (2015)). This decreases the computational costs for training and evaluating the network. In order to conserve the initial
input shape of the first dimension (time), we apply symmetric padding to minimise the introduction of artefacts related to the
155 borders.

While the original proposed concept of inception blocks has one max-pooling tower alongside the different convolution
stacks, we added a second pooling tower, which calculates the average on a kernel size of $3 \times 1$. Thus, one inception block
consists of three convolutional towers and two pooling (mean, and max) towers. A tower is defined as a collection or stack of
successive layers. The outputs of these towers are concatenated in the last layer of an inception block (see Fig. 3). Between
160 individual inception blocks, we added dropout layers (Srivastava et al., 2014) with a dropout rate of 0.35 to improve the
network's generalisation capability and prevent overfitting.

Moreover, we use batch normalisation layers (Ioffe and Szegedy, 2015) between each main convolution and activation layer
to accelerate the training process (Fig. 3). Those normalisations ensure that for each batch the mean activation is zero with
standard deviation of one. As proposed in Szegedy et al. (2015), the network has an additional minor tail which helps to
165 eliminate the vanishing gradient problem. Additionally, the minor tail helps to spread the internal representation of data as it
strongly penalises large errors.

The loss function for the main tail is the mean squared error:

$$L_{\mathrm{main}} = \frac{1}{n} \sum_i \left( y_{i,\mathrm{true}} - y_{i,\mathrm{pred}} \right)^2, \tag{2}$$

while the loss function of the minor tail is:

$$L_{\mathrm{minor}} = \frac{1}{n} \sum_i \left( |y_{i,\mathrm{true}} - y_{i,\mathrm{pred}}| \right)^4. \tag{3}$$

All activation functions are exponential linear units (ELU) (Clevert et al., 2016), only the final output activations are linear
(minor and main tail).

The network is built with Keras 2.2.4 (Chollet, 2015) and uses TensorFlow 1.13.1 (Martín Abadi et al., 2015) as backend.
We use Adam as optimiser and apply an initial learning rate of $10^{-4}$ (see also Sect. A4).
We train the model for 300 epochs on the HPC system 'Jülich Wizard for European Leadership Science' (JUWELS, Jülich
Supercomputing Centre (2019)) which is operated by the Jülich Supercomputing Centre (see also A3 for further details regard-
ing the software and hardware configurations).





## 4 Evaluation metrics and reference models

In general, one can interpret a supervised machine learning approach as an attempt to find an unknown function $\varphi$ which
maps an input pattern ($\mathbf{X}$) to the corresponding labels or the ground truth ($\boldsymbol{y}$). The machine learning model is consequently
an estimator ($\hat{\varphi}$) which maps $\mathbf{X}$ to an estimate $\hat{\boldsymbol{y}}$ of the ground truth. The goodness of the estimate is quantified by an error
function, which the network tries to minimise during training. As the network is only an estimator of the true function, the
mapping generally differs:

$$\varphi\left(\mathbf{X}\right) = \boldsymbol{y} \neq \hat{\boldsymbol{y}} = \hat{\varphi}\left(\mathbf{X}\right). \tag{4}$$

To evaluate the genuine added value of any meteorological or air quality forecasting model, it is essential to apply proper
statistical metrics. The following section describes the verification tools, which are used in this study.

### 4.1 Joint Distributions

Forecasts and observations are treated as random variables. Let $p(m,o)$ represent the joint distribution of a model's forecast $m$
and an observation $o$, which contains information on the forecast, the observation and the relationship between both of them
(Murphy and Winkler, 1987). To access specific pieces of information, we factorise the joint distribution into a conditional and
a marginal distribution in two ways. The first factorisation is called *calibration-refinement* and reads

$$p(m,o) = p(o|m)\,p(m), \tag{5}$$

where $p(o|m)$ is the conditional distribution of observing $o$ given the forecast $m$ and $p(m)$ is the marginal distribution which
indicates how often different forecast values are used (Murphy and Winkler, 1987; Wilks, 2006). A continuous forecast is
perfectly calibrated if

$$E\left(o|m\right) = m \tag{6}$$

holds, where $E\left(o|m\right)$ is the expected value of $o$ given the forecast $m$. The marginal distribution $p(m)$ is a measure of how often
different forecasts are made and is therefore also called refinement or sharpness. Both distributions are important to evaluate a
model's performance. Murphy and Winkler (1987) pointed out that a perfectly calibrated forecast is worth nothing if it lacks
refinement.

The second factorisation is called *likelihood-base rate* and consequently is given by

$$p(m,o) = p\left(m|o\right)p(o), \tag{7}$$

where $p\left(m|o\right)$ is the conditional distribution of forecast $m$ given that $o$ was observed. $p(o)$ is the marginal distribution which
only contains information about the underlying rate of occurrence of observed values and is therefore also called *sample*
*climatological distribution* (Wilks, 2006).





## 4.2 Scores and Skill Scores

To quantify a model's informational content, *scores* like the mean squared error (Eq. (8)) are defined to provide an absolute performance measure, while *skill scores* provide a relative performance measure related to a reference forecast (Eq. (9)).

$$MSE(\boldsymbol{m}, \boldsymbol{o}) = \frac{1}{N} \sum_{i=1}^{N} (m_i - o_i)^2 \geq 0, \tag{8}$$

Here, $N$ is the number of forecast-observation pairs, $\boldsymbol{m}$ is a vector containing all model forecasts, and $\boldsymbol{o}$ is a vector containing the observations (Murphy, 1988).

A skill score $S$ may be interpreted as the percentage of improvement of $A$ over the reference $A_{\text{ref}}$. Its general form is

$$S = \frac{A - A_{\text{ref}}}{A_{\text{perf}} - A_{\text{ref}}}. \tag{9}$$

Here, $A$ represents a general score, $A_{\text{ref}}$ is the reference score, and $A_{\text{perf}}$ the perfect score.

For $A = A_{\text{ref}}$ $S$ becomes zero, indicating that the forecast of interest has no improvements over the reference forecast. A value of $S > 0$ indicates an improvement over the reference, while $S < 0$ indicates a deterioration. The informative value of a skill score strongly depends on the selected reference forecast. In case of the mean squared error (Eq. (8)) the perfect score is equal to zero and Eq. (9) reduces to

$$S(\boldsymbol{m}, \boldsymbol{r}, \boldsymbol{o}) = 1 - \frac{MSE(\boldsymbol{m}, \boldsymbol{o})}{MSE(\boldsymbol{r}, \boldsymbol{o})}, \tag{10}$$

where $\boldsymbol{r}$ is a vector containing the reference forecast.

## 4.3 Reference models

We used three different reference models: persistence, climatology, and an ordinary least square model (linear regression). For the climatological reference we create four sub-reference models (see Sect. 4.3.2). In the following we introduce our basic reference models.

### 4.3.1 Persistence Model

One of the most straightforward model to build, which in general has good forecast skills on short lead times, is a persistence model. Today's observation of ozone dma8eu concentration is also the prediction for the next four days. Obviously, the skill of persistence decreases with increasing lead time. The good performance on short lead times is mainly due to the facts that weather conditions influencing ozone concentrations generally do not change rapidly, and that the chemical lifetime of ozone
is long enough.

### 4.3.2 Climatological reference models

We create four different climatological reference models (CASE I to CASE IV), which are based on the climatology of observations by following Murphy (1988) (also with respect to their terminology, which means that the reference score $A_{\text{ref}}$ is calculated by using the reference forecast $\boldsymbol{r}$).





The first reference forecast ($A_{\mathrm{ref}} : r = \overline{o}$, CASE I) is the internal single value climatology which is the mean of all observation during the test period. This unique value is then applied as reference for all forecasts. As this forecast has only one constant value which is the expectation value, this reference model is well calibrated but not refined at all.

The second reference ($A_{\mathrm{ref}} : r = o^*$, CASE II) is an internal multi-valued climatology. Here, we calculate twelve arithmetic means, where each of the means is the monthly mean over all years in the test set (e.g. one mean for all Januaries from 2012 to 2015, one for all Februaries, etc.). The corresponding monthly mean is applied as reference. Therefore, CASE II allows testing if the model has skill in reproducing the seasonal cycle of the observations.

The third reference ($A_{\mathrm{ref}} : r = \overline{\mu}$, CASE III) is an external single value climatology which is the mean of all available observations during the training and validation periods. This reference does not include any direct information on the test set. Therefore, one can access the information if the forecast of interest captures features which are not directly present in the train and validation set.

Finally, the fourth reference ($A_{\mathrm{ref}} : r = \mu^*$, CASE IV) is an external multi-valued climatology. A tabular summary explaining the individual formulae and terms following Murphy (1988) is given in Appendix A2. The last two references are calculated on a much longer time series than the first ones.

### 4.3.3 OLS reference model

The third reference model is an ordinary least square model. We train the OLS model by using the statsmodels package v0.10 (Seabold and Perktold, 2010). The OLS model is trained on the same data as the neural network (train set) and serves as a linear competitor.

## 5 Results

As described in Sect. 3 we split our data into three subsets (training, validation, and test set). We only used the independent test data set to evaluate the forecasting capabilities of the IntelliO3-ts network. During training and hyperparameter optimisation, only the train and validation sets were used, respectively. Therefore, the following results reflect the true generalisation capability of IntelliO3-ts. Before discussing the results in detail below, we would like to point out again, that this is the first time that one neural network has been trained to forecast data from an entire national air quality monitoring station network. Also, the network has been trained exclusively with time series data from air pollutant measurements and a meteorological reanalysis. No additional information, such as geographic coordinates, or other hints that could be used by the network to perform a station classification task, have been used. The impact of such extra information will be the subject of another study.

### 5.1 Forecasting results

Figure 4 shows the observed monthly $O_3$-dma8eu distribution (green) and the corresponding network predictions for a lead time of up to four days (dark to light blue) summarised for all 204 stations in the test set. The network clearly captures the seasonal cycle. Nonetheless, the arithmetic mean (black triangles) and the median tend to shift towards their respective annual



mean with increasing lead time (see also Fig. 7a to 7d). In spring and autumn, the observed and forecasted distributions match well, while in summer and wintertime the network underestimates the interquartile range (IQR) and does not reproduce the extremes values (for example, the maxima in July/August or the minima in December/January/February).

## 5.2 Comparison with competitive models

The skill scores based on the mean squared error (MSE) evaluated over all stations in the test set are summarised in Fig. 5. In the left and center groups of boxes and whiskers, the IntelliO3-ts model (labelled "CNN") and the OLS model are compared against persistence as reference. The right group of boxes and whiskers shows the comparison between IntelliO3-ts and OLS. The mean skill score for IntelliO3-ts against persistence is positive and increases with time. The OLS forecast shows similar behaviour in terms of its temporal evolution, but exhibits a slightly lower skill score throughout the 4-day forecasting period.

The increases in skill score in both cases is mainly due to the decreased score of the persistence model (see also Sect. 4.3.1). Consequently, IntelliO3-ts shows a positive skill score when the OLS model is used as a reference, indicating a small genuine added value over the OLS model.

In comparison with climatological reference forecasts as introduced in Sect. 4.2 and summarised in TableA2, the skill scores are high for the first lead time (1d) and decrease with increasing lead time. Both cases with a single value as reference (internal
CASE I, external CASE III) maintain a skill score above 0.4 over the four days. These high skill scores are a direct result of the fact that IntelliO3-ts captures the seasonal cycle as shown in Fig. 4, while the reference forecasts only report the overall mean as a single value prediction.

If the reference includes the seasonal variation (CASE II and CASE IV), the IntelliO3-ts skill score is still better than 0.4 for the first day (1d), but then it decreases rapidly and even becomes negative on day 4 for CASE II. The skill scores for CASE
II are lower than for CASE IV as the reference climatology (i.e. the monthly mean values) is calculated on the test set itself. These results show that, for the vast majority of stations, our model performs much better than a seasonal climatology for a one-day forecast, and it is still substantially better than the climatology after two days. However, there are some stations, which yield a negative skill score even on day 2 in the CASE II comparison. Longer-term forecasts with this model set-up do not add value compared to the computationally much cheaper monthly mean climatological forecast.

## 5.3 Analysis of joint distributions

The full joint distribution in terms of calibration refinement factorisation (Sect. 4.1) is shown in Fig. 7a (first lead time; 1d) to Fig. 7d (last lead time; 4d). The marginal distribution (refinement) is shown as histogram (light grey; sample size), while the conditional distribution (calibration) is presented by specific percentiles in different line styles. If the median (.5th quantile, solid line) is below the reference, the network exhibits a high-bias with respect to the observations and vice versa.
Obviously, quantiles in value regions with many data samples are more robust and therefore credible than quantiles in data-sparse concentration regimes (Murphy et al., 1989). On the first lead time (d1, Fig. 7a), the IntelliO3-ts network has a tendency to slightly over-predict concentrations $\lesssim 30\,\mathrm{ppb}$. On the other hand, the network under-forecasts concentrations above $\gtrsim 70\,\mathrm{ppb}$.





Both, very high, and very low forecasts are rare (note the logarithmic axis for the sample size). Therefore, the results in these
regimes have to be treated with caution. Further detail is provided in Fig. A1, where the conditional biases are shown (terms
BI, BII, BIV in Sect. A2). and decrease the maximal climatological potential skill score (term AI, see also TableA2).

With increasing lead time the model looses its capability to predict concentrations close to zero and high concentrations
above $80\,\mathrm{ppb}$. The marginal distribution develops a pronounced bimodal shape which is directly linked to the conditional
biases. The number of high (extreme) ozone concentrations is relatively low, resulting in few training examples. The network
tries to optimise the loss function with respect to the most common values. As a result, predictions of concentrations near
the mean value of the distribution are generally more correct than predictions of values from the fringes of the distribution.
Moreover, this also explains, why the model does not perform substantially better than the monthly mean climatology forecasts
(CASE II, CASE IV). This problem also becomes apparent in other studies. For example, Sayeed et al. (2020) focus their
categorical analysis on a threshold value of $55\,\mathrm{ppbv}$ (maximum 8h-average) which corresponds to the air quality index value
'moderate' (AQI 51 to 100), instead of the legal threshold value of $70\,\mathrm{ppbv}$ (U.S. Environmental Protection Agency, 2016,
Table 5, therein), as the model shows better skills in this regime.

To shed more light on the factors influencing the forecast quality, we analyse the network performance individually for
each season (DJF, MAM, JJA, and SON). Conditional quantile plots for individual seasons can be found in the supplementary
material (A5). As mentioned above, the bimodal shape of the marginal distribution is mainly caused by the network's weakness
to predict very high and low ozone concentrations. Moreover, the seasonal decomposition shows that the left mode is caused
by the fall (SON) and winter (DJF) seasons (Fig. Fig. A6a to A6d and A3a to A3d). In both seasons, the most common
values fall into the same concentration range, while the right tail of SON is much more pronounced than for DJF with higher
values occurring primarily in September. In the summer season (JJA, Fig. A5a to A5d) the most frequently predicted values
correspond to the location of the right mode of Fig. 7a to 7a. During DJF, MAM, and JJA the model has a stronger tendency of
under-forecasting with increasing lead time (median line moves above the reference line).

## 5.4 Relevance of input variables

To analyse the impact of individual input variables on the forecast results, we apply a bootstrapping technique as follows: we
take the original input of one station, keep eight of the nine variables unaltered, and randomly draw (with replacement) the
missing variable (20 times per variable per station). This destroys the temporal structure of this specific variable so that the
network will no longer be able to use this information for forecasting. Compared to alternative approaches, such as re-training
the model with fewer input variables, setting all variable values to zero, etc., this method has two main advantages: (i) the
model does not need to be re-trained and thus the evaluation occurs with the exact same weights that were learned from the full
dataset, and (ii) the distribution of the input variable remains unchanged so that adverse effects, for example due to correlated
input variables, are excluded. However, we note, that this method may underestimate the impact of a specific variable in case
of correlated input data. Also, this analyses only evaluates the behaviour of the deep learning model and does not evaluate the
impact of these variables on actual ozone formation in the atmosphere.





After the randomisation of one variable, we apply the trained model on this modified input data and compare the new prediction with the original one. For comparison, we apply the skill score (Eq. (9)) based on the MSE where we use the original forecast as reference. Consequently, the skill score will be negative if the bootstrapped variable has a significant
impact on model performance. Figure 8 shows the skill scores for all variables (x-axis) and lead times (dark (1d) to light blue (4d) boxplots). Ozone is the most crucial input variable, as it has by far the lowest skill score for all lead times. With increasing lead time, the skill score increases but stays lower than for any other variable. In contrast, the model does not derive much skill from the variables nitrogen oxide, nitrogen dioxide, and the planetary boundary layer height. In other words, the network does not perform worse, when randomly drawn values replace one of those original time series. Relative humidity, temperature and
the wind's u-component have an impact on the model performance. With increasing lead time, these influences decrease.

## 6   Conclusions

In this study, we developed and evaluated IntelliO3-ts, a deep learning forecasting model for daily near-surface ozone concentrations (dma8eu) at arbitrary air quality monitoring stations in Germany. The model uses chemical (O3, NO, NO2) and meteorological time-series of the previous six days to create forecasts for up to four days into the future. IntelliO3-ts is based
on convolutional inception blocks, which allow to calculate concurrent convolutions with different kernel sizes. The model has been trained on 10 years of data from 313 background stations in Germany. Hyperparameter tuning and model evaluation were done with independent data sets of 2 and 6 years length, respectively.

The model generalises well and generates good quality forecasts for lead times up to two days. These forecasts are superior compared to the reference models persistence, ordinary least squares, annual and seasonal climatology. After 2 days, the
350 forecast quality degrades, and the forecast adds no value compared to a monthly mean climatology of dma8eu ozone levels. We could primarily attribute this to the network's tendency to converge to the mean monthly value. The model does not have any spatial context information which could counteract this tendency. Near-surface ozone concentrations at background stations are highly influenced by air mass advection, but the IntelliO3-ts network yet has no way to take upwind information into account. We will investigate spatial context approaches in a forthcoming study.

We observed, that the model loses refinement with increasing lead time which results in unsatisfactory predictions on the tails of the observed ozone concentration. We were able to attribute this weakness to the under-representation of extreme (either very small or high) levels in the training data set. This is a general problem for machine learning applications and regression methods. The machine learning community is investigating possible solutions to lessen the impact of such data imbalances, but their adaptation is beyond the scope of this paper as proposed techniques are not directly applicable to those time series
(auto-correlation time).

Bootstrapping individual time series of the input data to analyse the importance of those variables on the predictive skill showed, that the model mainly focused on the previous ozone concentrations. Temperature and relative humidity only have a small effect on the model performance, while the time series of NO, $NO_2$, and PBL have no impact.





The IntelliO3-ts network extends previous work by using a new network architecture, and training one model on a much

larger set of measurement station data and longer time periods. In light of Rasp and Lerch (2018) who used several neural networks to postprocess ensemble weather forecasts, we applied meteorological evaluation metices to perform a point-by-point comparison, which is not common in the field of deep learning. We hope that the forecast quality of IntelliO3-ts can be further improved if we take spatial context information into account so that the advection of background ozone and ozone precursors can be learned by the model.

*Code and data availability.* The current version of IntelliO3-ts is available from the project website: https://gitlab.version.fz-juelich.de/toar/ machinelearningtools/-/tree/IntelliO3-ts-v1.0_initial-submit under the MIT licence (http://opensource.org/licenses/mit-license.php). The exact version of the model and data used to produce the results in this paper are archived on b2share (Kleinert et al., 2020).

**Appendix A**

**A1   Information on used stations**

Table A1 lists all measurement stations which we used in this study. The table also shows the number of samples $(\mathbf{X}, \boldsymbol{y})$ for each of the three data sets (train, validation, test)

**A2   Mean Squared Error Decomposition (Murphy, 1988)**

This section provides additional information about the MSE decomposition introduced by Murphy (1988). The MSE decomposition is performed as

$$380 \quad MSE\left(\boldsymbol{m},\boldsymbol{o}\right) = \frac{1}{n}\sum_{i=1}^{n}\left(\left(m_i - \overline{m}\right) - \left(o_i - \overline{o}\right) + \left(\overline{m} - \overline{o}\right)\right)^2 \tag{A1}$$

$$= \left(\overline{m} - \overline{o}\right)^2 + \sigma_m^2 + \sigma_o^2 - 2\sigma_{mo} \tag{A2}$$

$$= \left(\overline{m} - \overline{o}\right)^2 + \sigma_m^2 + \sigma_o^2 - 2\sigma_m\sigma_o\rho_{mo}. \tag{A3}$$

Here $\sigma_m$ $(\sigma_o)$ is the sample variance of the forecasts (observations) and $\sigma_{mo}$ is the sample covariance of the forecasts and observations, which is given by $\sigma_{mo} = \frac{1}{n}\sum_{i=1}^{n}\left(m_i - \overline{m}\right)\left(o_i - \overline{o}\right)$. $\rho_{mo}$ is the sample coefficient of correlation between forecast

and observation.





CASE I:

$$S(m,\overline{o},o) =$$

$$\underbrace{\rho_{mo}^2}_{\text{AI}} - \underbrace{\left(\rho_{mo} - \frac{\sigma_m}{\sigma_o}\right)^2}_{\text{BI}} - \underbrace{\left(\frac{\overline{m} - \overline{o}}{\sigma_o}\right)^2}_{\text{CI}} \tag{A4}$$

CASE II:

$$S(m,o^\star,o) =$$

$$\frac{\text{AI} - \text{BI} - \text{CI} - \rho_{o^\star o}^2 + \left(\rho_{o^\star o} - \frac{\sigma_{o^\star}}{\sigma_o}\right)^2}{\underbrace{1 - \rho_{o^\star o}^2}_{\text{AII}} + \underbrace{\left(\rho_{o^\star o} - \frac{\sigma_{o^\star}}{\sigma_o}\right)^2}_{\text{BII}}} \tag{A5}$$

CASE III:

$$S(m,\overline{\mu},o) =$$

$$\frac{\text{AI} - \text{BI} - \text{CI} + \left(\frac{\overline{\mu} - \overline{o}}{\sigma_o}\right)^2}{\underbrace{1 + \left(\frac{\overline{\mu} - \overline{o}}{\sigma_o}\right)^2}_{\text{AIII}}}. \tag{A6}$$

CASE IV:

$$S(m,\mu,o) =$$

$$\frac{\text{AI} - \text{BI} - \text{CI} - \rho_{\mu o}^2 + \left(\rho_{\mu o}^2 - \frac{\sigma_\mu}{\sigma_o}\right)^2 + \left(\frac{\overline{\mu} - \overline{o}}{\sigma_o}\right)^2}{\underbrace{1 - \rho_{\mu o}^2}_{\text{AIV}} + \underbrace{\left(\rho_{\mu o}^2 - \frac{\sigma_\mu}{\sigma_o}\right)^2}_{\text{BIV}} + \underbrace{\left(\frac{\overline{\mu} - \overline{o}}{\sigma_o}\right)^2}_{\text{CIV}}}. \tag{A7}$$

The term AI is the square of the sample correlation coefficient and might be interpreted as the strength of linear relationship between the forecast and the observation. This term ranges from zero (no correlation) to one (perfect correlation). Term BI includes the square of the differences between the sample correlation coefficient and the ratio of standard deviation of the forecast and observation. Therefore, BI is a measure of the conditional bias of the forecast which is always positive due to the

405 square and tends to decrease skill as it is a subtrahend. The last term, which is included in all cases I-IV, is CI and contains the square of the difference of the mean forecast and mean observation divided by the variance of the observation. Therefore, CI is





a measure of the unconditional bias in the forecast and, again, tends to decrease the skill as it is a subtrahend which is always greater or equal to zero.

In case of multi-value internal climatology (Case II Eq. (A5)), two additional terms appear in the dominator as well as the numerator which tend to decrease skill in general and only vanish if $2\rho_{o^\star o} = \sigma_\mu/\sigma_o$. In Case III, the additional term AIII appears that includes the square of the difference between the mean external and the internal climatology divided by the variance of the observation. AIII leads to an increase of skill for any difference in the means of external and internal climatologies.

Three additional terms (AIV, BIV and CIV) appear if Eq. (10) is decomposed by using a multi-valued external climatology as reference forecast (Case IV). These terms only vanish if $2\rho_{\mu o} = \sigma_\mu/\sigma_o$ and $\overline{\mu} = \overline{o}$. A summary of all four cases and all terms included is given in TableA2

Figure A1 also includes all individual terms as described above.

## A3  Additional information on JUWELS

Each node on JUWELS (Jülich Supercomputing Centre, 2019) which is part of the graphical processor unit (GPU) partition is equipped with four NVIDIA Volta V100 GPUs. The user guide for JUWELS is available from https://apps.fz-juelich.de/jsc/hps/juwels/index.html.

## A4  Detailed model settings

Figure A2 shows the full architecture of IntelliO3-ts including all individual layers and tails. Table A3 lists the specific compile options per keyword of keras' complile method. Table A4 summarises additional settings for the specific architecture

## A5  Seasonal decomposition of conditional quantiles

The following section contains all conditional quantile plots decompoed for all seasons (DJF: Fig. A3a to Fig. A3d, MAM: Fig. A4a to Fig. A4d, JJA: Fig. A5a to Fig. A5d, and SON: Fig. A6a to Fig. A6d)

*Author contributions.* FK and MGS developed the concept of the study. All authors jointly developed the concept of the machine learning model. FK implemented the neural network and performed the experiment. FK had the lead in writing the manuscript with contributions from LHL and MGS. LHL had the technical lead in code development and workflow design. All authors revised the final manuscript and accepted to submit to GMD.

*Competing interests.* The authors declare that they have no conflict of interest.



*Acknowledgements.* We are thankful to all air quality data providers which made their data available in the TOAR database. Moreover, we thank the meteorological section of the Institute of Geosciences at the University of Bonn, which provided the COSMO Reanalysis data. We
thank Sabine Schröder for the help to access data through the JOIN interface and Jenia Jitsev for helpful discussions.

The authors gratefully acknowledge the Gauss Centre for Supercomputing e.V. (www.gauss-centre.eu) for funding this project by providing computing time through the John von Neumann Institute for Computing (NIC) on the GCS Supercomputer JUWELS at Jülich Supercomputing Centre (JSC).





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





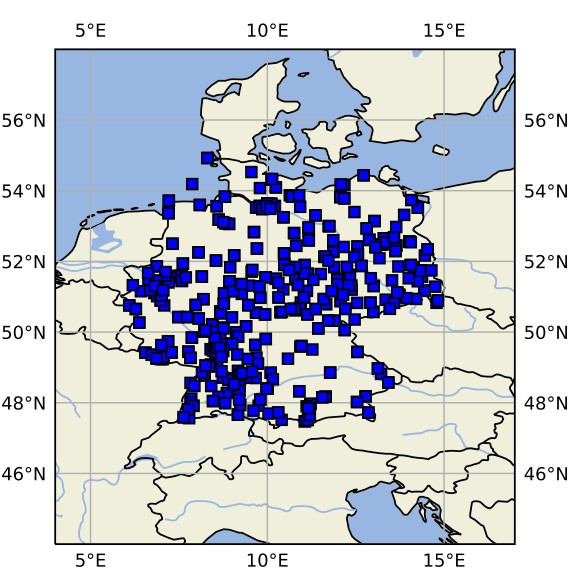

**Figure 1.** Map of central Europe showing the location of German measurement sites used in this study.



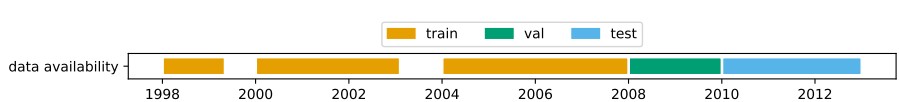

**Figure 2.** Data availability diagram combined for all variables and all stations. The training set is coloured in orange, the validation set in green, and the test set in blue. Gaps in 1999 and 2003 are caused by missing model data in the TOAR database.



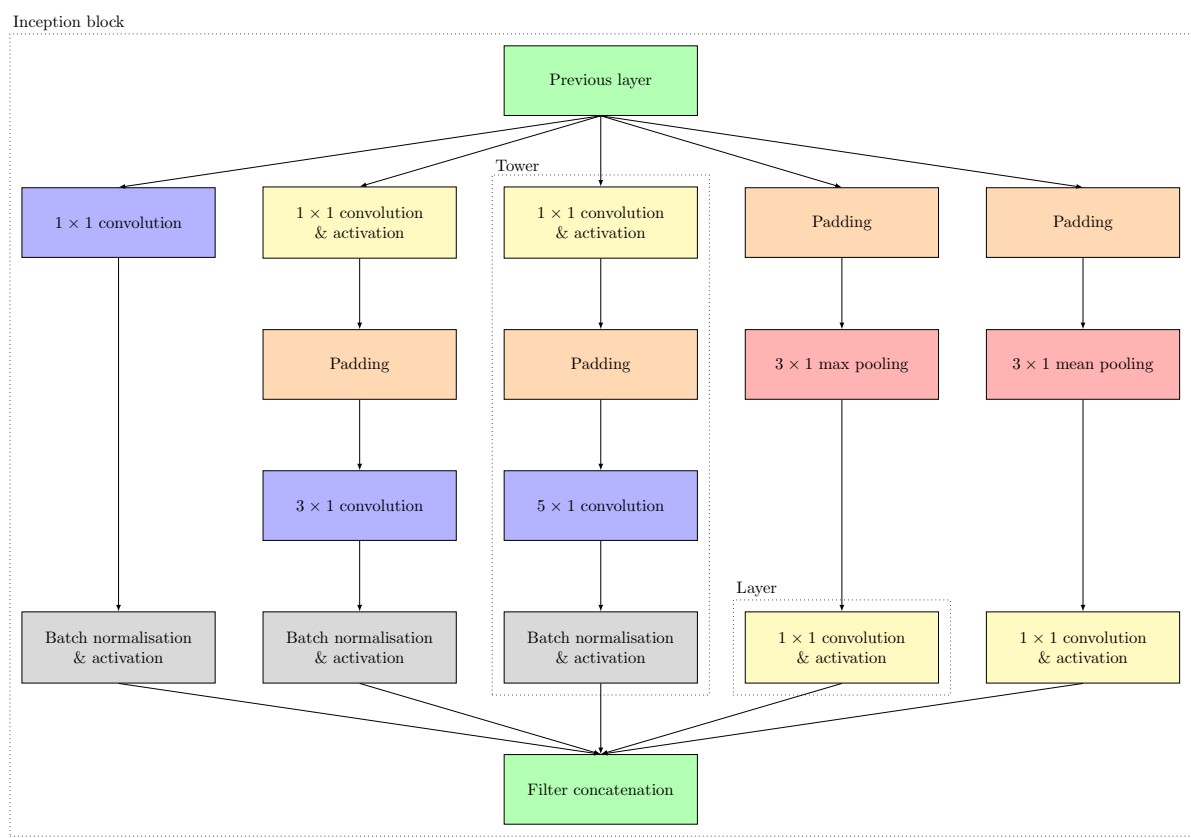

**Figure 3.** Inception block with dimensionality reduction and batch normalisation before activation. Successive layers are grouped together as a tower. The input and output layers are colour coded in green, pooling layers in red, filter reduction layers in yellow, padding layers in orange, a tower's main convolution in blue, and batch normalisation including activation in grey (extended version of Szegedy et al. (2015, Fig. 2(b)))

.



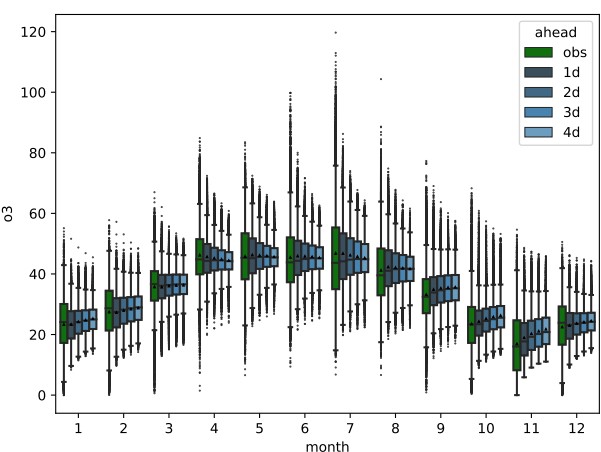

**Figure 4.** Monthly dma8eu ozone concentrations for all test-stations as boxplots. Measurements are denoted by "orig" (green), while the forecasts are denoted by "1d" (dark blue) to "4d" (light blue). Whiskers have a maximal length of one interquartile range. The black triangles denote the arithmetic means.



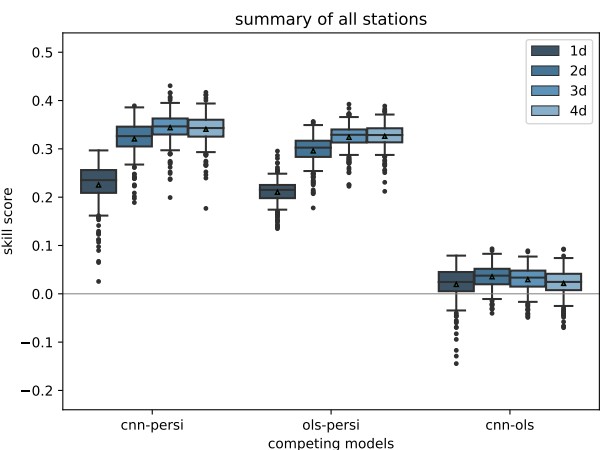

**Figure 5.** Skill scores of the IntelliO3-ts (cnn) versus the two reference models persistence (persi) and, ordinary least square (ols) based on the mean squared error; separated for all lead times (1d (dark blue) to 4d (light blue)). Positive values denote that the first mentioned prediction model performs better than the reference model (mentioned as second). The triangles denote the arithmetic means.



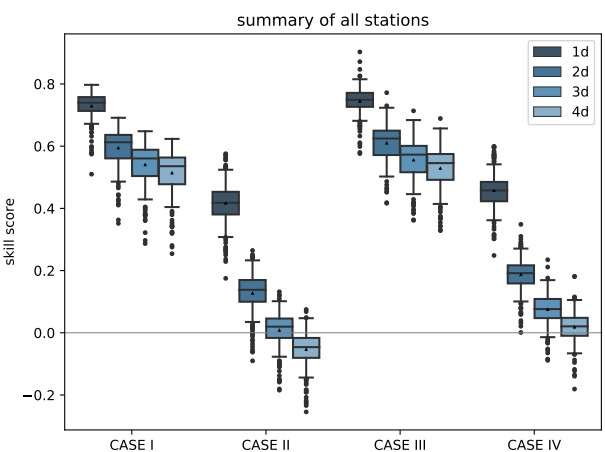

**Figure 6.** Skill scores of IntelliO3-ts with respect to climatological reference forecasts: with internal single value reference (CASE I), internal multi value (monthly) reference (CASE II), external single (CASE III) and external multi (monthly) reference (CASE IV) for all lead times from 1d (dark blue) to 4d (light blue). Triangles denote the arithmetic means.





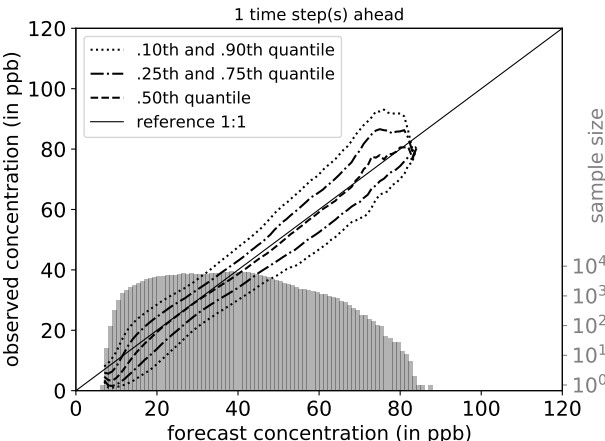

**Figure 7a.** Conditional quantile plot for all IntelliO3-ts predictions for a lead time of one day (1d). Conditional percentiles (.10th and .90th, .25th and .75th and .50th) from the conditional distribution $f(o_j|m_i)$ are shown as lines in different styles. The reference line indicates a hypothetic perfect forecast. The marginal distribution of the forecast $f(m_i)$ is shown as log-histogram (right axis, light grey). All calculations are done by using a bin size of 1ppb. Quantiles are smoothed by using a rolling mean of window size three. (Original design by Murphy et al. (1989))



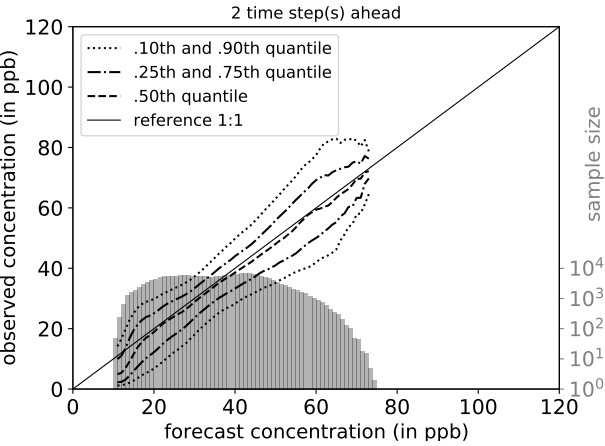

**Figure 7b.** Same as Fig. 7a but for a lead time of two days (2d)



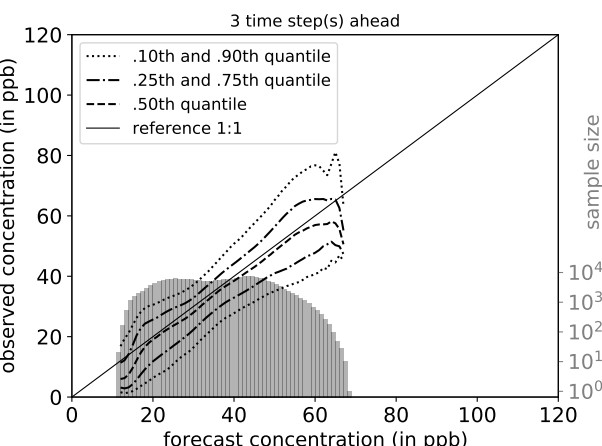

**Figure 7c.** Same as Fig. 7a but for a lead time of three days (3d)



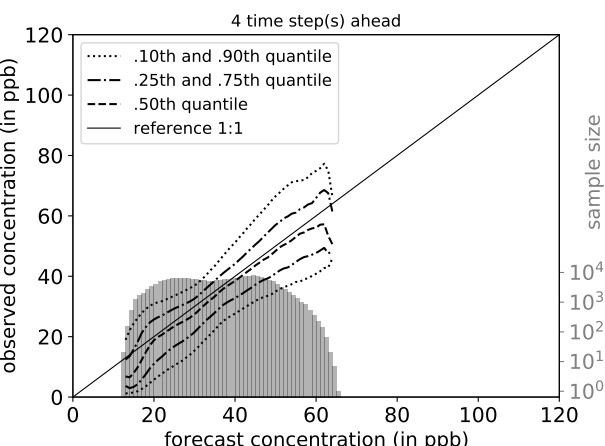

**Figure 7d.** Same as Fig. 7a but for a lead time of four days (4d)



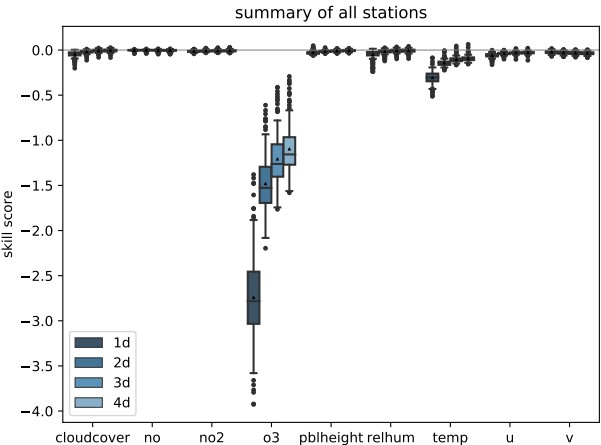

**Figure 8.** Skill scores of bootstrapped model predictions having the original forecast as the reference model are shown as boxplots for all lead times from 1d (dark blue) to 4d (light blue).



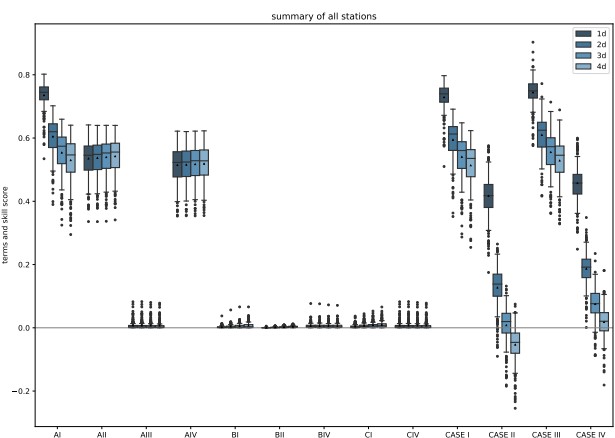

**Figure A1.** Skill scores of IntelliO3-ts with respect to climatological reference forecast; with internal single value reference (CASE I), internal multi value (monthly) reference (CASE II), external single (CASE III) and external multi (monthly) reference (CASE IV) for all lead times from 1d (dark blue) to 4d (light blue). All terms are described in Sect. A2 Triangles denote the arithmetic means.





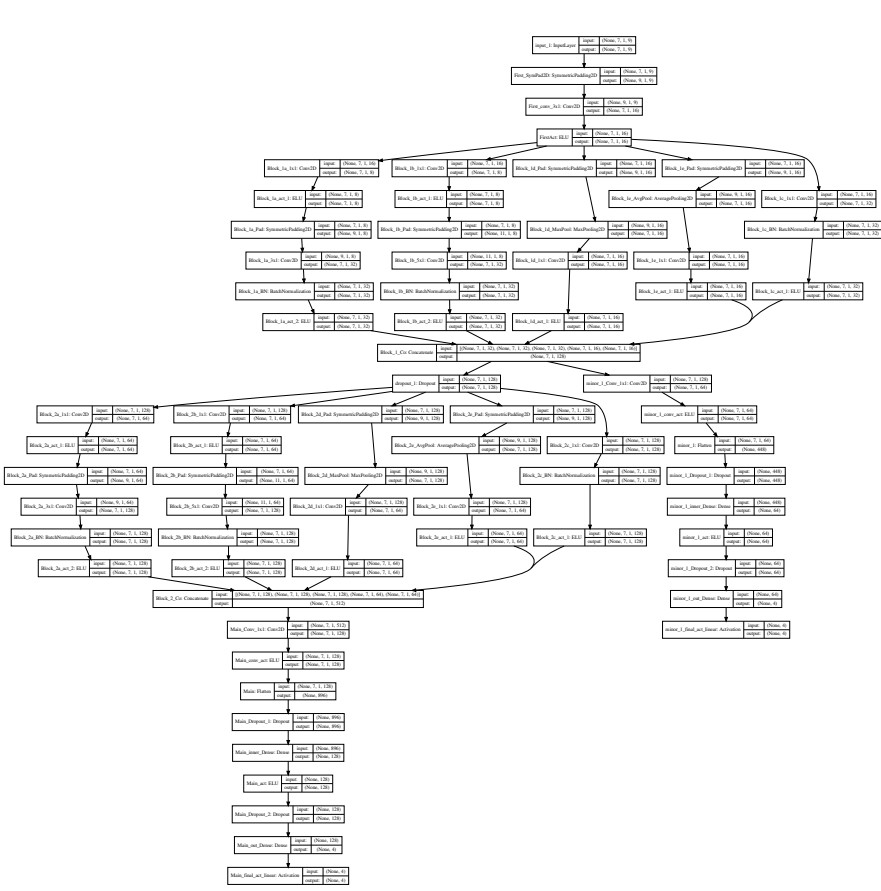

**Figure A2.** Network architecture containing all inception blocks, heads, and the additional average pooling layer.





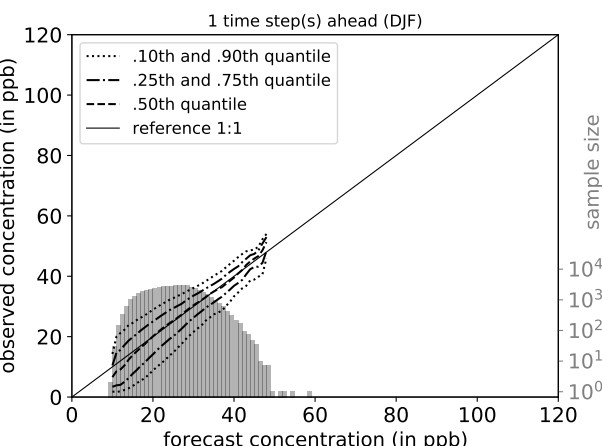

**Figure A3a.** Same as Fig. 7a but for DJF



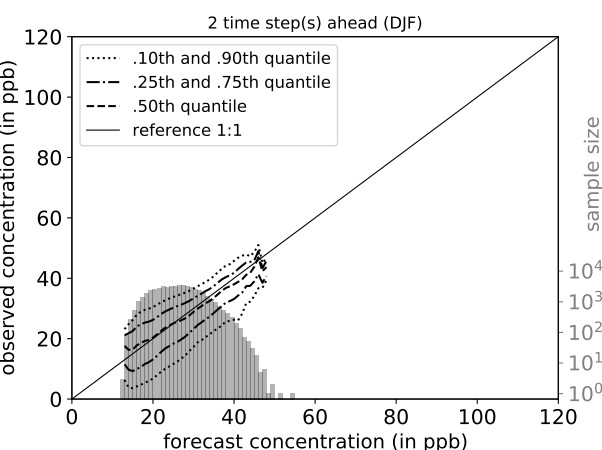

**Figure A3b.** Same as Fig. A3a but for a lead time of two days (2d)





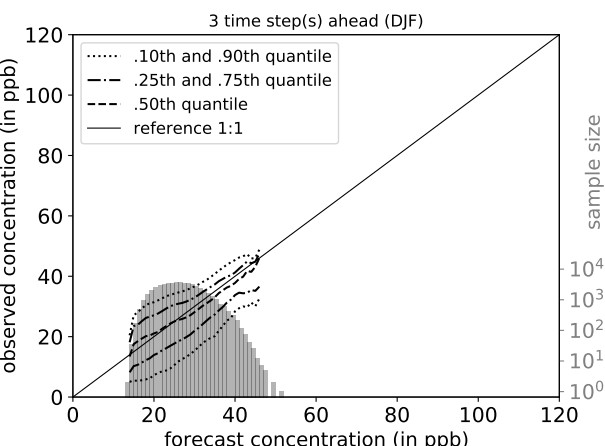

**Figure A3c.** Same as Fig. A3a but for a lead time of three days (3d)



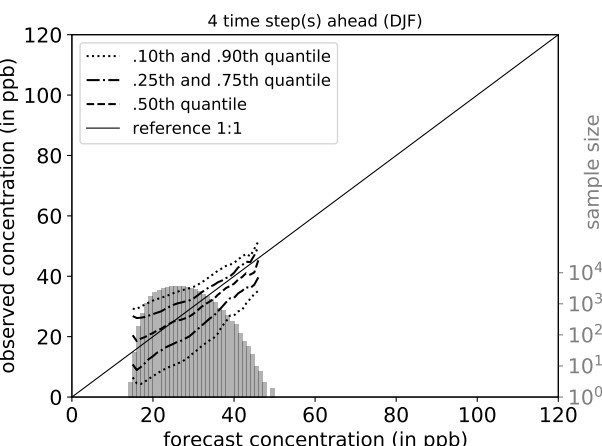

**Figure A3d.** Same as Fig. A3a but for a lead time of four days (4d)





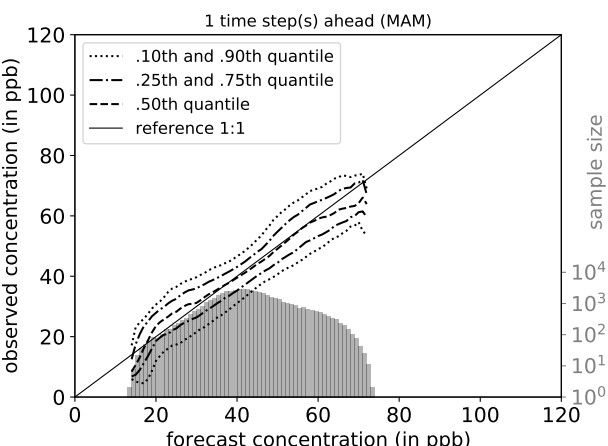

**Figure A4a.** Same as Fig. 7a but for MAM





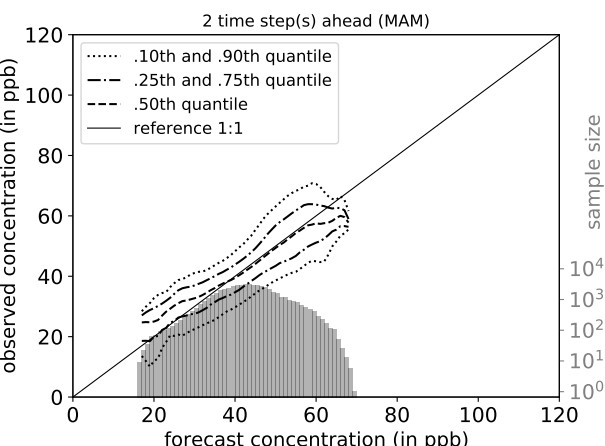

**Figure A4b.** Same as Fig. A4a but for a lead time of two days (2d)



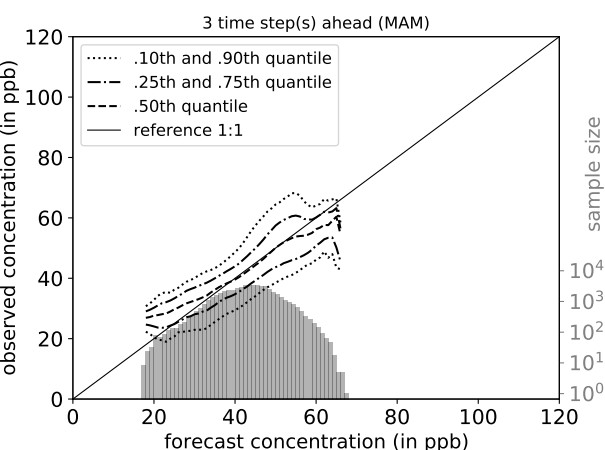

**Figure A4c.** Same as Fig. A4a but for a lead time of three days (3d)





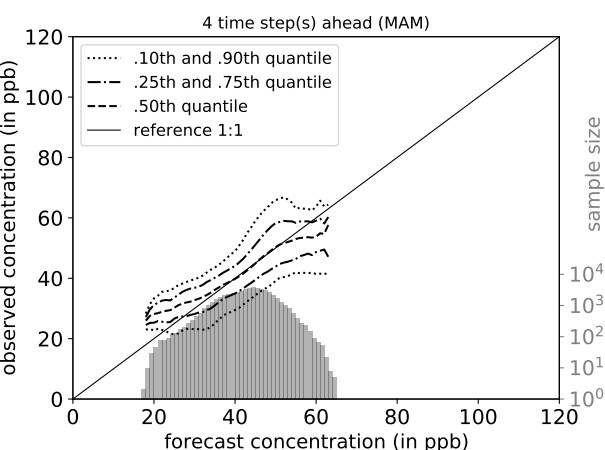

**Figure A4d.** Same as Fig. A4a but for a lead time of four days (4d)




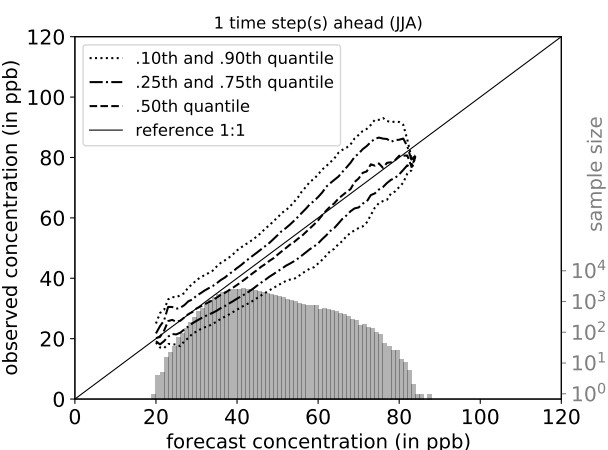

**Figure A5a.** Same as Fig. 7a but for JJA





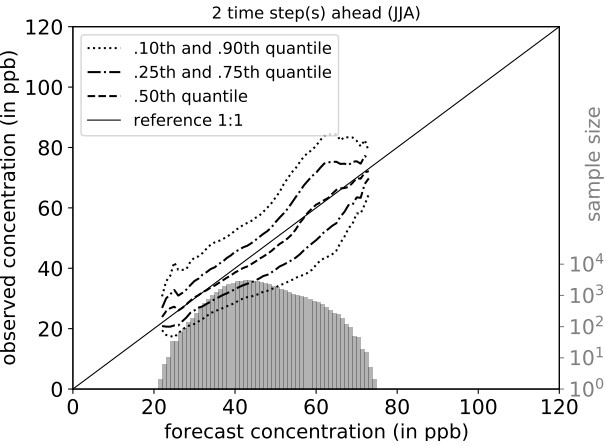

**Figure A5b.** Same as Fig. A5a but for a lead time of two days (2d)



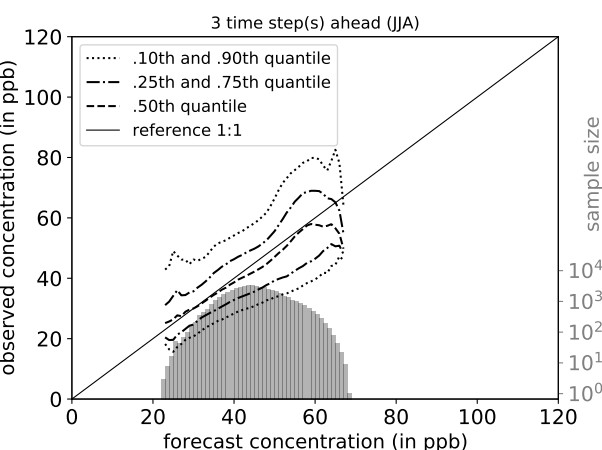

**Figure A5c.** Same as Fig. A5a but for a lead time of three days (3d)





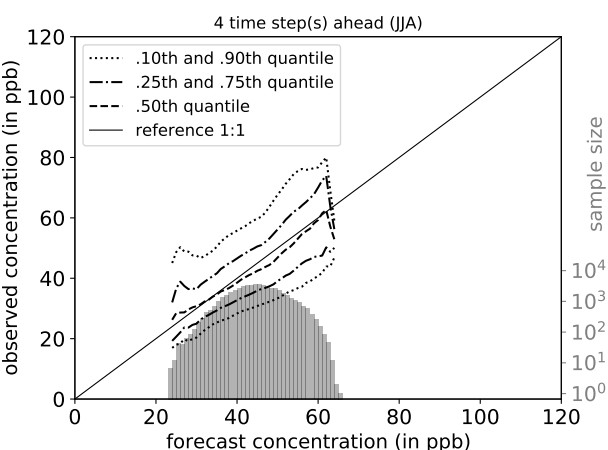

**Figure A5d.** Same as Fig. A5a but for a lead time of four days (4d)





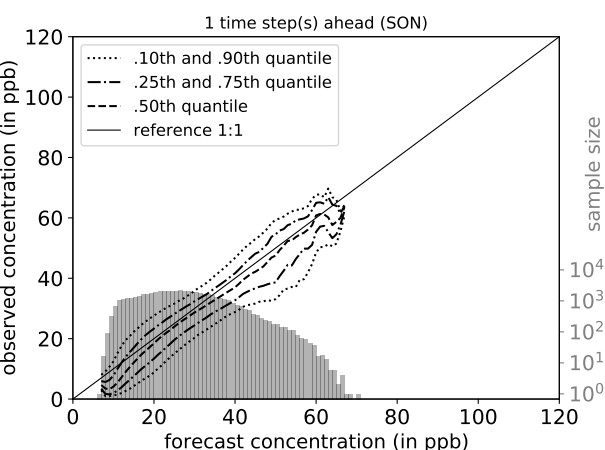

**Figure A6a.** Same as Fig. 7a but for SON





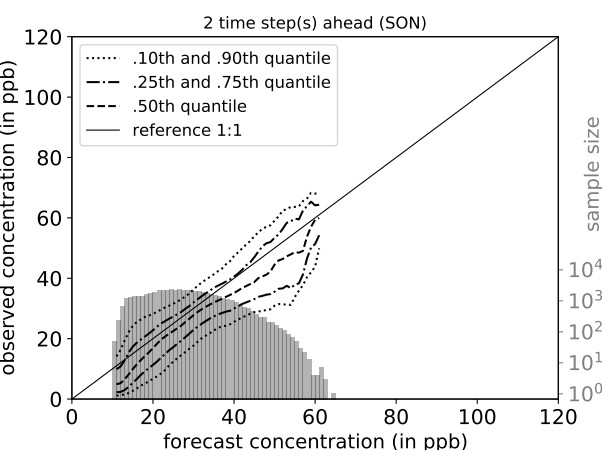

**Figure A6b.** Same as Fig. A6a but for a lead time of two days (2d)





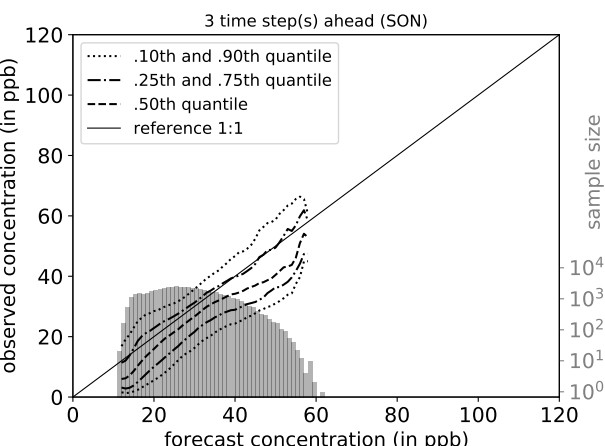

**Figure A6c.** Same as Fig. A6a but for a lead time of three days (3d)



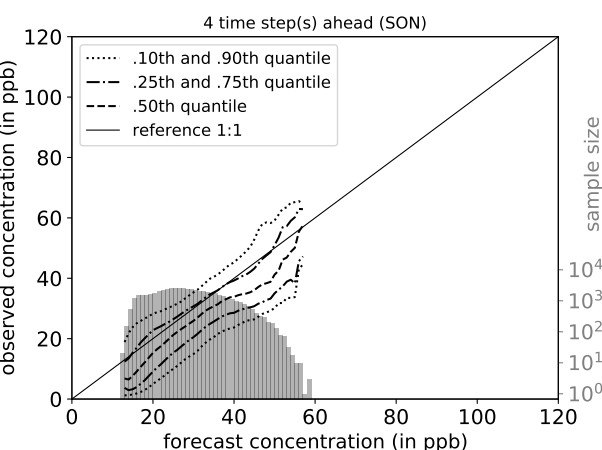

**Figure A6d.** Same as Fig. A6a but for a lead time of four days (4d)



**Table 1.** Overview of the literature on ozone forecasts with neural networks. Machine learning (ML) types are abbreviated as FC for fully connected, CNN for convolutional neural networks, RNN for recurrent neural networks, and LSTM for long-short term memories. We use the following abbreviations for time periods : yr for years and m for month.

| Citation | ML type | Total number of stations | Stations for training | Time period | comments |
|---|---|---|---|---|---|
| Comrie (1997) | FC | 8 | 8 | 5yr | random split for train, val |
| Cobourn et al. (2000) | FC | 7 | 7 | 5yr (train)+ 1yr (val)+1yr(test) | |
| Prybutok et al. (2000) | FC | 1 | | 4m + 1m | |
| Gardner and Dorling (2001) | FC | 6 | 6 | 12yr | |
| Eslami et al. (2019) | CNN | 25 | 25 | 3yr train 1yr test | random split |
| Liu et al. (2019) | attention RNN | 2 | 2 | 10m (train), 7 days (test) | Analysis for $PM_{2.5}$ |
| Maleki et al. (2019) | FC | 4 | 4 | 1yr | random split for train, val, test |
| Silva et al. (2019) | FC | 2 | 1 | 13yr (train, val test); 14yr (test on second station) | dry deposition of $O_3$; random split on first station |
| Abdul Aziz et al. (2019) | FC | 1 | 1 | 7days | individual measurements for study |
| Pawlak and Jarosławski (2019) | FC | 2 | 2 | 6m (train) + 6m (test) | Individual network per station |
| Ma et al. (2020) | Bidir.-LSTM | 19 (standard) + 48 (transfer) | 19 (standard) + 48 (transfer) | 9m | exploration of transfer learning |
| Sayeed et al. (2020) | CNN | 21 | 21 | 3yr (train) + 1yr (test / retrain) | retrain model for each prediction |
| Zhang et al. (2020) | CNN-LSTM | 35 | 35 | 19m | Gridded forecast |
| this study | inception blocks | 329 | 313 | 18 yr | |





**Table 2.** Input variables and applied daily statistics according to Table 3.

| Variable | daily statistics |
|---|---|
| NO | dma8eu |
| $NO_2$ | dma8eu |
| $O_3$ | dma8eu |
| cloudcover | average |
| planetary boundary layer height | maximum |
| relative humidity | average |
| temperature | maximum |
| wind's u-component | average |
| wind's v-component | average |





**Table 3.** Definitions of statistical metrics in TOAR analysis relevant for this study. Adopted form Schultz et al. (2017, Supplement 1, Table 6, therein)

| Name | Description |
| --- | --- |
| data_capture | Fraction of valid (hourly) values available in the aggregation period. |
| average_values | Daily [...] average value. No data capture criterion is applied, i.e. a daily average is valid if at least one hourly value of the day is present. |
| dma8eu | As dma8epa, but using the EU definition of the daily 8-hour window starting from 17 h of the previous day. (dma8epa: Daily maximum 8-hour average statistics according to the US EPA definition. 8-hour averages are calculated for 24 bins starting at 0 h local time. The 8-h running mean for a particular hour is calculated on the concentration for that hour plus the following 7 hours. If less than 75% of data are present (i.e. less than 6 hours), the average is considered missing. Note that in contrast to the official EPA definition, a daily value is considered valid if at least one 8-hour average is valid.) |
| maximum | Daily maximum value. No data capture criterion is applied, i.e. a daily maximum is valid if at least one hourly value of the day is present. |





**Table 4.** Number of stations and total number of samples (pairs of $\mathbf{X}$ and $\boldsymbol{y}$) used in the training (train), validation (val), and test (test) data sets, respectively. The number of stations per set varies as not all stations have data through the full period (see Table A1 for details).

|                    | train  | val    | test   |
| ------------------ | ------ | ------ | ------ |
| Number of Stations | 313    | 212    | 204    |
| Number of Samples  | 643788 | 145030 | 212093 |

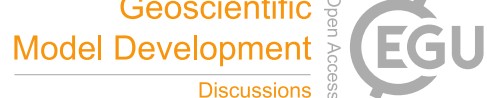

**Table A1.** Number of samples (input and output pairs) per station separated by training (train), validation (val), and test data set. "—" denotes no samples in a set.

| | train | val | test |
|---|---|---|---|
| stat. ID | | | |
| DEBB001 | 1104 | — | — |
| DEBB006 | 1455 | — | — |
| DEBB007 | — | 721 | 1086 |
| DEBB009 | 1438 | — | — |
| DEBB021 | 2512 | 705 | 1052 |
| DEBB024 | 2592 | — | — |
| DEBB028 | 1353 | — | — |
| DEBB031 | 2577 | — | — |
| DEBB036 | 1008 | — | — |
| DEBB038 | 1245 | — | — |
| DEBB040 | 760 | — | — |
| DEBB042 | 2902 | 721 | 1086 |
| DEBB043 | 2194 | — | — |
| DEBB048 | 2473 | 721 | 1075 |
| DEBB050 | 2510 | 721 | — |
| DEBB051 | 1006 | — | — |
| DEBB053 | 2115 | 706 | 1086 |
| DEBB055 | 1887 | 721 | 1053 |
| DEBB063 | 1392 | 721 | 1086 |
| DEBB064 | 1480 | 721 | 1086 |
| DEBB065 | 1411 | 699 | 1086 |
| DEBB066 | 1451 | 721 | 1086 |
| DEBB067 | 1073 | 721 | 1086 |
| DEBB075 | — | 721 | 1079 |
| DEBB082 | — | 622 | 1086 |
| DEBB083 | — | — | 1086 |
| DEBE010 | 1372 | 707 | 1035 |
| DEBE032 | 2441 | 690 | 1060 |
| DEBE034 | 2506 | 671 | 1031 |
| DEBE051 | 2433 | 694 | 1053 |





|  | train | val | test |
|---|---|---|---|
| stat. ID |  |  |  |
| DEBE056 | 2481 | 678 | 1055 |
| DEBE062 | 1941 | 677 | 1069 |
| DEBW004 | 1440 | 721 | 1086 |
| DEBW006 | 1451 | 721 | 1086 |
| DEBW007 | 1440 | 699 | — |
| DEBW008 | 656 | — | — |
| DEBW010 | 3041 | 708 | 1086 |
| DEBW013 | 1432 | 710 | 1079 |
| DEBW019 | 2962 | 710 | 1078 |
| DEBW020 | 1520 | — | — |
| DEBW021 | 1579 | — | — |
| DEBW023 | 1430 | 721 | 1078 |
| DEBW024 | 3011 | 713 | 1086 |
| DEBW025 | 1530 | — | — |
| DEBW026 | 3005 | 721 | — |
| DEBW027 | 3005 | 699 | 1075 |
| DEBW028 | 1510 | — | — |
| DEBW029 | 3012 | 721 | 1086 |
| DEBW030 | 2966 | — | — |
| DEBW031 | 2970 | 711 | 1069 |
| DEBW032 | 2648 | — | — |
| DEBW034 | 3045 | 707 | — |
| DEBW035 | 2281 | — | — |
| DEBW036 | 1183 | — | — |
| DEBW037 | 3023 | 721 | — |
| DEBW039 | 2999 | 710 | 1086 |
| DEBW041 | 1581 | — | — |
| DEBW042 | 2617 | 708 | 1067 |
| DEBW044 | 1571 | — | — |
| DEBW045 | 656 | — | — |





|          | train | val | test |
|----------|-------|-----|------|
| stat. ID |       |     |      |
| DEBW046  | 2990  | 699 | 1086 |
| DEBW047  | 1563  | —   | —    |
| DEBW049  | 646   | —   | —    |
| DEBW050  | 1556  | —   | —    |
| DEBW052  | 2652  | 721 | 1078 |
| DEBW053  | 1574  | —   | —    |
| DEBW054  | 1566  | —   | —    |
| DEBW056  | 2938  | 721 | 1086 |
| DEBW057  | 644   | —   | —    |
| DEBW059  | 2974  | 721 | 1086 |
| DEBW060  | 1571  | —   | —    |
| DEBW065  | 1540  | —   | —    |
| DEBW072  | 480   | —   | —    |
| DEBW076  | 3035  | 721 | 708  |
| DEBW081  | 2654  | 721 | 1079 |
| DEBW084  | 1444  | 721 | 1042 |
| DEBW087  | 3043  | 713 | 1086 |
| DEBW094  | 2188  | —   | —    |
| DEBW102  | 1122  | —   | —    |
| DEBW103  | 2525  | 721 | —    |
| DEBW107  | 1801  | 714 | 1086 |
| DEBW110  | 1107  | 721 | —    |
| DEBW111  | 1083  | 703 | —    |
| DEBW112  | 651   | 721 | 1079 |
| DEBW113  | 678   | —   | —    |
| DEBY002  | 2924  | 721 | 747  |
| DEBY004  | 2917  | 707 | 1028 |
| DEBY005  | 2959  | 714 | 1086 |
| DEBY013  | 1412  | 652 | 980  |
| DEBY017  | 1250  | —   | —    |



|  | train | val | test |
|---|---|---|---|
| stat. ID |  |  |  |
| DEBY020 | 2976 | 721 | 1013 |
| DEBY031 | 2927 | 678 | 1072 |
| DEBY032 | 2975 | 721 | 711 |
| DEBY034 | 1555 | — | — |
| DEBY039 | 2554 | 721 | 1067 |
| DEBY047 | 1895 | 721 | 754 |
| DEBY049 | 2918 | 693 | 1066 |
| DEBY052 | 2929 | 708 | 1035 |
| DEBY062 | 1411 | 704 | 748 |
| DEBY072 | 2907 | 690 | 1055 |
| DEBY077 | 1409 | 721 | 724 |
| DEBY079 | 2878 | 721 | 671 |
| DEBY081 | 2932 | 523 | 713 |
| DEBY082 | 1592 | — | — |
| DEBY088 | 2986 | 713 | 1062 |
| DEBY089 | 2644 | 721 | 1086 |
| DEBY092 | 616 | — | — |
| DEBY099 | 1828 | 703 | 725 |
| DEBY109 | 1310 | 713 | 1071 |
| DEBY113 | 1347 | 706 | 1086 |
| DEBY118 | 937 | 705 | 727 |
| DEBY122 | — | — | 877 |
| DEHB001 | 2567 | 710 | 953 |
| DEHB002 | 2287 | 702 | 1053 |
| DEHB003 | 2546 | 695 | — |
| DEHB004 | 1428 | 708 | 1037 |
| DEHB005 | 2518 | 683 | 1066 |
| DEHE001 | 1451 | 721 | 1086 |
| DEHE008 | 2447 | 707 | 1075 |
| DEHE010 | 1600 | — | — |




|  | train | val | test |
|---|---|---|---|
| stat. ID |  |  |  |
| DEHE013 | — | 721 | 1086 |
| DEHE017 | 1562 | — | — |
| DEHE018 | 3016 | 721 | 1086 |
| DEHE019 | 1958 | — | — |
| DEHE022 | 2643 | 721 | 1086 |
| DEHE023 | 2966 | 708 | 466 |
| DEHE024 | 2935 | 710 | 1086 |
| DEHE025 | 1554 | — | — |
| DEHE026 | 2877 | 697 | 1086 |
| DEHE027 | 1536 | — | — |
| DEHE028 | 2946 | 710 | 1068 |
| DEHE030 | 3028 | 721 | 1086 |
| DEHE032 | 2926 | 714 | 1075 |
| DEHE033 | 1835 | — | — |
| DEHE034 | 1880 | — | — |
| DEHE039 | — | — | 812 |
| DEHE042 | 2966 | 721 | 1079 |
| DEHE043 | 3004 | 721 | 1074 |
| DEHE044 | 2543 | 721 | 1086 |
| DEHE045 | 2535 | 699 | 1086 |
| DEHE046 | 2513 | 714 | 1086 |
| DEHE048 | 1043 | — | — |
| DEHE050 | 1014 | — | — |
| DEHE051 | 2331 | 721 | 1086 |
| DEHE052 | 2078 | 713 | 1086 |
| DEHE058 | 789 | 721 | 1086 |
| DEHE060 | 704 | 672 | 1086 |
| DEHH008 | 1439 | 721 | 1086 |
| DEHH021 | 2624 | 710 | 1086 |
| DEHH033 | 2148 | 682 | 1067 |





|  | train | val | test |
| --- | --- | --- | --- |
| stat. ID |  |  |  |
| DEHH047 | 2175 | 696 | 1086 |
| DEHH049 | 2150 | 721 | 1075 |
| DEHH050 | 2131 | 721 | 1069 |
| DEMV001 | 794 | — | — |
| DEMV004 | 2908 | 721 | 1058 |
| DEMV007 | 2986 | 721 | 1053 |
| DEMV012 | 2885 | 710 | 1086 |
| DEMV017 | 2507 | 708 | 1086 |
| DEMV018 | 2113 | 710 | — |
| DEMV019 | 1429 | 706 | 1086 |
| DEMV021 | 600 | 688 | 1072 |
| DEMV024 | — | — | 908 |
| DENI011 | 2611 | 452 | 1086 |
| DENI016 | 3034 | 627 | 1051 |
| DENI019 | 2919 | — | — |
| DENI020 | 2984 | 667 | 1086 |
| DENI028 | 2927 | 516 | 1086 |
| DENI029 | 2903 | 692 | 1086 |
| DENI031 | 1410 | 451 | 1079 |
| DENI038 | 2599 | 573 | 1083 |
| DENI041 | 2935 | 525 | 1086 |
| DENI042 | 2939 | 553 | 1072 |
| DENI043 | 2941 | 606 | 1086 |
| DENI051 | 2976 | — | 1072 |
| DENI052 | 2910 | 529 | 1086 |
| DENI054 | 2997 | 596 | 1086 |
| DENI058 | 2398 | — | 1086 |
| DENI059 | 2408 | 451 | 1079 |
| DENI060 | 2386 | 677 | 1086 |
| DENI062 | 2482 | 625 | 1080 |





|  | train | val | test |
|---|---|---|---|
| stat. ID |  |  |  |
| DENI063 | 2385 | 460 | 1073 |
| DENI077 | — | — | 1079 |
| DENW004 | 1148 | — | — |
| DENW006 | 1367 | 694 | 1026 |
| DENW008 | 2511 | 701 | 1064 |
| DENW010 | 1397 | — | — |
| DENW013 | 1830 | — | — |
| DENW015 | 1451 | — | — |
| DENW018 | 1196 | — | — |
| DENW028 | 1655 | — | — |
| DENW029 | 2530 | — | — |
| DENW030 | 2785 | 630 | 998 |
| DENW036 | 1314 | — | — |
| DENW038 | 2598 | 652 | 1079 |
| DENW042 | 1185 | — | — |
| DENW047 | 1350 | — | — |
| DENW050 | 2488 | — | — |
| DENW051 | 1206 | — | — |
| DENW053 | 1795 | 678 | 1051 |
| DENW059 | 1777 | 648 | 980 |
| DENW062 | 1078 | — | — |
| DENW063 | 2816 | — | — |
| DENW064 | 2887 | 589 | 1052 |
| DENW065 | 2877 | 550 | 1045 |
| DENW066 | 2865 | — | — |
| DENW067 | 2473 | 686 | 1079 |
| DENW068 | 2892 | 447 | 1009 |
| DENW071 | 1827 | 713 | 1071 |
| DENW078 | 1382 | 681 | 1078 |
| DENW079 | 2040 | 706 | 1086 |





|  | train | val | test |
| --- | --- | --- | --- |
| stat. ID |  |  |  |
| DENW080 | 2147 | 689 | 1043 |
| DENW081 | 2422 | 646 | 1058 |
| DENW094 | 1980 | 627 | 1058 |
| DENW095 | 1981 | 681 | 1071 |
| DENW096 | 700 | — | — |
| DENW179 | 766 | 699 | 1079 |
| DENW247 | — | 572 | 1066 |
| DERP001 | 1421 | 721 | 1068 |
| DERP007 | 2652 | 721 | 1077 |
| DERP013 | 2883 | 708 | 1067 |
| DERP014 | 2967 | 703 | 1061 |
| DERP015 | 2810 | 710 | 1047 |
| DERP016 | 2962 | 721 | 1086 |
| DERP017 | 2955 | 721 | 1055 |
| DERP019 | 1413 | 708 | 1041 |
| DERP021 | 2996 | 713 | 1025 |
| DERP022 | 2989 | 701 | 1059 |
| DERP025 | 2918 | 691 | 1086 |
| DERP028 | 2802 | 678 | 1026 |
| DESH005 | 962 | — | — |
| DESH006 | 614 | — | — |
| DESH008 | 3031 | 721 | 1053 |
| DESH016 | 2635 | 698 | — |
| DESH021 | 1107 | — | — |
| DESH023 | 1700 | 721 | 1066 |
| DESH033 | — | 721 | 1072 |
| DESL003 | 1393 | 713 | 1086 |
| DESL008 | 1371 | — | — |
| DESL011 | 2776 | 710 | 1086 |
| DESL012 | — | — | 1086 |





|            | train | val | test |
|------------|-------|-----|------|
| stat. ID   |       |     |      |
| DESL017    | 2785  | 714 | 1086 |
| DESL018    | 1656  | 710 | 1086 |
| DESL019    | 1371  | 472 | 1057 |
| DESN001    | 2925  | 689 | 1072 |
| DESN004    | 3011  | 705 | 1086 |
| DESN005    | 1166  | —   | —    |
| DESN011    | 2613  | 694 | 1073 |
| DESN012    | 2995  | 721 | —    |
| DESN014    | 2256  | —   | —    |
| DESN017    | 3028  | 704 | —    |
| DESN019    | 2934  | 721 | —    |
| DESN024    | 3017  | 721 | —    |
| DESN028    | 401   | —   | —    |
| DESN036    | 810   | —   | —    |
| DESN045    | 2913  | 721 | 1064 |
| DESN050    | 2939  | 710 | —    |
| DESN051    | 1451  | 685 | 1075 |
| DESN057    | 1535  | —   | —    |
| DESN059    | 2533  | 714 | 1078 |
| DESN074    | 2534  | 702 | 1062 |
| DESN076    | 2489  | 717 | 1075 |
| DESN079    | —     | —   | 1071 |
| DESN085    | 713   | —   | —    |
| DESN092    | —     | 536 | 1057 |
| DEST002    | 3020  | 721 | 1075 |
| DEST005    | 802   | —   | —    |
| DEST011    | 2924  | 713 | 1075 |
| DEST014    | 996   | —   | —    |
| DEST022    | 805   | —   | —    |
| DEST025    | 480   | —   | —    |





|  | train | val | test |
|---|---|---|---|
| stat. ID |  |  |  |
| DEST028 | 2694 | — | — |
| DEST030 | 2282 | — | — |
| DEST031 | 796 | — | — |
| DEST032 | 447 | — | — |
| DEST039 | 2971 | 676 | 1069 |
| DEST044 | 2923 | 709 | 1086 |
| DEST050 | 2672 | 706 | 1075 |
| DEST052 | 1484 | — | — |
| DEST061 | 814 | — | — |
| DEST063 | 1241 | — | — |
| DEST066 | 2991 | 659 | 1086 |
| DEST069 | 2589 | 707 | — |
| DEST070 | 1488 | — | — |
| DEST071 | 462 | — | — |
| DEST072 | 2641 | 703 | — |
| DEST077 | 1611 | 663 | 1086 |
| DEST078 | 3042 | 709 | — |
| DEST089 | 2467 | 710 | 1048 |
| DEST098 | 1422 | 649 | 1086 |
| DEST104 | — | — | 1061 |
| DETH005 | 3030 | 721 | 1075 |
| DETH009 | 2995 | 721 | 1086 |
| DETH013 | 2945 | 710 | 1086 |
| DETH016 | 1937 | — | — |
| DETH018 | 3027 | 721 | 1086 |
| DETH020 | 3002 | 721 | 1086 |
| DETH024 | 1193 | — | — |
| DETH025 | 2542 | 370 | — |
| DETH026 | 1474 | 721 | 1072 |
| DETH027 | 1444 | 705 | 1086 |





|  | train | val | test |
| --- | --- | --- | --- |
| stat. ID |  |  |  |
| DETH036 | 2996 | 721 | 1086 |
| DETH040 | 2926 | 721 | 1078 |
| DETH041 | 3003 | 710 | 1043 |
| DETH042 | 2993 | 697 | 1086 |
| DETH060 | 2519 | 721 | 1086 |
| DETH061 | 2465 | 721 | 1063 |
| DETH095 | — | — | 1059 |
| DETH096 | — | — | 898 |
| DEUB001 | 1202 | 721 | 1075 |
| DEUB003 | 1436 | — | — |
| DEUB004 | 2746 | 721 | 932 |
| DEUB005 | 1422 | 721 | 974 |
| DEUB013 | 414 | — | — |
| DEUB021 | 369 | — | — |
| DEUB026 | 1768 | — | — |
| DEUB028 | 2602 | 603 | 1086 |
| DEUB029 | 2834 | 721 | 1062 |
| DEUB030 | 2893 | 710 | 947 |
| DEUB031 | 1845 | — | — |
| DEUB032 | 1629 | — | — |
| DEUB033 | 2034 | — | — |
| DEUB034 | 1434 | — | — |
| DEUB035 | 1977 | — | — |
| DEUB036 | 411 | — | — |
| DEUB038 | 1628 | — | — |
| DEUB039 | 1676 | — | — |
| DEUB040 | 1549 | — | — |
| DEUB041 | 781 | — | — |
| DEUB042 | 687 | — | — |
| Tot. stations | 313 | 212 | 204 |
| Tot. samples | 643788 | 145030 | 212093 |





**Table A2.** Summarised skill scores $S(\boldsymbol{m}, \boldsymbol{r}, \boldsymbol{o})$ based on the MSE (CASE I - IV) and relating terms (AI - CIV) as described in Sect. A2. $\boldsymbol{m}$ denotes the prediction model, $\boldsymbol{r}$ is the reference and $\boldsymbol{o}$ denotes the observation. The '$\times$' sign marks if a term (AI to CIV) appears in the different factorisations. The following abbreviations are used: corr. for correlation, obs. for observation, ref. reference (forecast), cond. for conditional, int. for internal, and ext. for external.

| Term | CASE I | CASE II | CASE III | CASE IV | Formula | Meaning |
|---|---|---|---|---|---|---|
| $\boldsymbol{r}$: | $\overline{o}$ | $o^{\star}$ | $\overline{\mu}$ | $\boldsymbol{\mu}$ | | reference |
| | int. single | int. multi | ext. single | ext. multi | | |
| AI | $\times$ | $\times$ | $\times$ | $\times$ | $\rho_{mo}^2$ | potential skill |
| AII | | $\times$ | | | $\rho_{o^{\star}o}^2$ | corr. obs.-ref. |
| AIII | | | $\times$ | | $\left(\frac{\overline{\mu}-\overline{o}}{\sigma_o}\right)^2$ | trend |
| AIV | | | | $\times$ | $\rho_{\mu o}^2$ | corr. obs.-ref. |
| BI | $\times$ | $\times$ | $\times$ | $\times$ | $\left(\rho_{mo}-\frac{\sigma_m}{\sigma_o}\right)^2$ | cond. bias |
| BII | | $\times$ | | | $\left(\rho_{o^{\star}o}-\frac{\sigma_{o^{\star}}}{\sigma_o}\right)^2$ | cond. bias |
| BIV | | | | $\times$ | $\left(\rho_{\mu o}^2-\frac{\sigma_\mu}{\sigma_o}\right)^2$ | cond. bias |
| CI | $\times$ | $\times$ | $\times$ | $\times$ | $\left(\frac{\overline{m}-\overline{o}}{\sigma_o}\right)^2$ | uncond. bias |
| CIV | | | | $\times$ | $\left(\frac{\overline{\mu}-\overline{o}}{\sigma_o}\right)^2$ | trend |
| Eq. | (A4) | (A5) | (A6) | (A7) | | |





**Table A3.** Specific compile options passed to keras' compile method. Other keywords which are not listed in this table are left with default values.

| keyword | value |
| --- | --- |
| optimizer | adam(lr=0.001, amsgrad=True) |
| loss | Eq. (3), Eq. (2) |
| loss_weight | 0.01, 0.99 |





**Table A4.** Specific information and rates used to setup the model architecture

| Setting | value |
|---|---|
| dropout rate | 0.35 |
| reguralizer | keras.regularizers.l2(0.01) |
| epochs | 300 |
| activation (all without last layer) | ELU |
| activation (output layers) | linear |
| padding | symmetric padding |