# Peer review of "IntelliO3-ts v1.0: A neural network approach to predict near-surface ozone concentrations in Germany"

_Geoscientific Model Development, 2020_

## Referee Comment (RC1) · Anonymous Referee #1 · 10 Sep 2020

Kleinert et al. present a machine learning (ML) method to predict surface ozone concentrations up to four days in advance. The method uses convolutional neural networks (CNN) trained on an extensive set of historical data (10 years) to forecast the daily maximum 8-hour average ozone concentration at more than 300 background measurement sites across Germany. Based on only a few input variables (concentrations of ozone and nitrogen oxides (NOx) and six meteorological variables), the ML ozone forecasts show good skills for the first two days but don't perform better than reference forecasts over longer time windows. This is a very nice paper that is well written and easy to follow. Minor comments are given below.

My only major comment is the issue with trends in the input data. Presumably, the 10-year training data of ozone and NOx – and possibly temperature – show a long-term

trend? I would expect this to create issues for the CNN since this trend is imprinted in the training data (even after normalizing around the interannual mean and standard deviation). Further, given the long-term trends in both ozone and NOx, the test samples (2010-2015) might represent a different 'environmental regime' that the CNN was not trained on. The authors should discuss this in the revised version of the manuscript.

Minor comments:

- Table 1: I suggest you include the study by Seltzer et al. (2020).

- Section 2.1.: (Variable selection): using the daily maximum 8-hour average for NO and NO2 seems like an odd choice to me. From a chemical perspective, one would rather want to use the 24-hour average or maximum one-hour concentration?

- Section 4.1 (Joint Distributions): While interesting it's not clear why this section is in the manuscript. It doesn't seem to have much relevance for understanding the paper?

- Section 5.2. (Comparison with competitive models): please add reference to Figure 6.

References:

Seltzer, K. M., Shindell, D. T., Kasibhatla, P., and Malley, C. S.: Magnitude, trends, and impacts of ambient long-term ozone exposure in the United States from 2000 to 2015, Atmos. Chem. Phys., 20, 1757–1775, https://doi.org/10.5194/acp-20-1757-2020, 2020.

---

## Referee Comment (RC2) · Anonymous Referee #2 · 22 Sep 2020

The authors present a data-driven forecast model for maximum daily 8-hour ozone (mda8O3) concentrations based on multiple convolutional neural layers and apply it to the network of rural ozone monitoring sites in Germany. The manuscript is well written, and the model presented and its application make a valuable contribution to the field. I ask the authors for a few clarifications and specifications detailed in my specific comments below.

Specific comments:

1) The authors include several meteorological variables as input but not radiation. This is a little odd given the importance of radiation for photochemistry and thus ozone formation. Cloud-cover is used by the authors together with temperature as a proxy, however I am not surprised that this surrogate variable shows limited influence on

mda8O3. It seems also that direct and diffuse irradiance are available at various time steps (https://reanalysis.meteo.uni-bonn.de/?Download_Data___COSMO-REA6) although otherwise stated on page p4, L93.

2) Related to my previous comment I am a bit puzzled that all meteorological covariates show such limited added skill. Have the authors assessed simple brute-force correlations between meteorological covariates and mda8O3, are they as low as the skill score would suggest? Also I wonder if the limited influence stems from the joint consideration of all seasons and if a cleaner picture would emerge on seasonal or monthly basis.

3) Along these lines, I am wondering if pooling of observations in the samples might cause some spurious effects. Given the relatively low VOC abundances during fall-early spring pooling might explain to some degree the relative low skill obtained for NO and NO2.

4) On p.5., L103 the authors state that they include stations if they have at least one year of valid data in one of the sets. I wonder if an unequal inclusion of observations from different time periods affects the robustness of the training and tuning. How much would the sample reduce if a more stringent criterion would be applied say e.g., more than 80% or 50% coverage over the considered time period, or a high fraction of available data per month, season and year?

5) Also O3 and NO and NO2 show substantial changes and trends in Europe over the time period considered. The authors do not address this in their manuscript, thus I assume the data has not been detrended before use in model training and validation? What magnitude of effect would we expect by considering non-stationary time series training of a CNN model?

6) The authors use dma8 for NO and NO2. What is the motivation to use here dma8 instead of the daily mean or maximum value, which would be the more common quantities?

[Figure]

7) P6, L135, a batch size of 512 samples is used, how do the authors derive this number?

Minor comments:

1) I suggest to use 'training set' instead of 'train set' throughout the manuscript.

2) I suggest grouping several figures to multi-panel figures to increase accessibility (7a-d, A3a-d, A4a-d,A5a-d, A6a-d).

3) Figure A2 is incredibly hard to read even when zooming in to 400%.

4) Axis labels of Figure A2 are hard to read.

5) P7, L174 a reference for the Adam optimizer is missing

6) I was wondering if the section on joint distributions and skill scores could not be moved to the Appendix

Spelling and typos:

P6, L118: hyperparameters

P9L226: replace 'model with 'models'

P10, L235: replace 'observation' with 'observations'

P10, L237: replace 'multi-valued' with 'multi-value' and check throughout the text

P11, L295: therefore more credible

P11, 297: replace 'the network under-forecasts' with 'the forecast is underestimating'
* * *

---

## Author Comment (AC1) · 6 Nov 2020

**1    General statement**

We are grateful to the anonymous reviewers for the constructive review and encouraging comments. In the following we are going to address all comments point by point, while using blue for comments of Reviewer #1 and red for Reviewer #2. Additional changes not explicitly stated by the reviewers, are highlighted in magenta. We used the same colours for changes in the manuscript.

**2   Answer to Anonymous Referee #1**

"Kleinert et al. present a machine learning (ML) method to predict surface ozone concentrations up to four days in advance. The method uses convolutional neural networks(CNN) trained on an extensive set of historical data (10 years) to forecast the daily maximum 8-hour average ozone concentration at more than 300 background measurement sites across Germany. Based on only a few input variables (concentrations of ozone and nitrogen oxides (NOx) and six meteorological variables), the ML ozone forecasts show good skills for the first two days but don't perform better than reference forecasts over longer time windows. This is a very nice paper that is well written and easy to follow. Minor comments are given below."

- "My only major comment is the issue with trends in the input data. Presumably, the 10-year training data of ozone and NOx – and possibly temperature – show a long-term trend? I would expect this to create issues for the CNN since this trend is imprinted in the training data (even after normalizing around the interannual mean and standard deviation). Further, given the long-term trends in both ozone and NOx, the test samples (2010-2015) might represent a different 'environmental regime' that the CNN was not trained on. The authors should discuss this in the revised version of the manuscript."
  Reviewer #1 points to a very crucial point here. Schultz et al. (2020, accepted) suggest multiple different techniques to generate the training, validation and test sets, respectively, which might be subject of a separate study or even better of a model intercomparison study. We added the following paragraph (focusing on the applied data split) as a new section *Limitations and additional remarks*.
  By splitting the data into three consecutive, non-overlapping sets, we ensure that the data sets are as independent as possible. On the other hand, this independence comes at the cost, that changes of trends in the input variables may not be captured, especially as our input data are not de-trended. Indeed, at European

non-urban measurement sites, several ozone metrics related to high concentrations (e.g. 4th highest daily maximum 8-hour (4MDA8) or the $95\%$-percentile of hourly concentrations) show a significant decrease during our study period (1997 to 2015) (Fleming et al., 2018; Yan et al., 2018). Our data splitting method for evaluating the generalisation capability is conservative in the sense that we evaluate the model on the test set, which has the largest possible distance to the training set. If our research model shall be transformed into an operational system we suggest to apply online learning and use the latest available data for subsequent training cycles (see for example Sayeed et al. (2020)).
(Same answer to rev #2)

- "Minor comments":

  – "Table 1: I suggest you include the study by Seltzer et al. (2020)"
    We added Seltzer et al. (2020) and the following sentences on page 3 line 52: "Seltzer et al. (2020) used 3557 measurement sites across six measurement networks to analyse long term ozone exposure trends in North America by applying a fully connected neural network. They mainly focused on metrics related to human health and crop loss."

  – "Section 2.1.: (Variable selection): using the daily maximum 8-hour average for NO and NO2 seems like an odd choice to me. From a chemical perspective, one would rather want to use the 24-hour average or maximum one-hour concentration?"
    While we agree with the reviewer that the choice of metrics for NO and NO2 may seem strange, this was indeed done on purpose. We now added the following text to explain our rationale for this decision:
    "The choice of using the dma8eu metrics for NO and NO2 was motivated by the idea to sample all chemical quantities during the same time periods and with similar averaging times. While the dma8eu metrics is calculated based on data starting at 5pm on the previous day, the daily mean/max values
for example would be calculated based on data starting from the current day. To test the impact of using different metrics for ozone precursors we also trained the model from scratch with either mean or maximum concentrations of NO and NO2 as inputs. The results of these runs were hardly distinguishable from the results presented below."
(Same answer to rev #2)

– "Section 4.1 (Joint Distributions): While interesting it's not clear why this section is in the manuscript. It doesn't seem to have much relevance for understanding the paper?"
We agree, that this section is not mandatory to understand the figures of joint distributions. We therefore moved this subsection to the appendix as we also explicitly refer to the calibration refinement factorisation in other sections.

– "Section 5.2. (Comparison with competitive models): please add reference to Figure 6."
We added the reference to Fig. 6

"References: Seltzer, K. M., Shindell, D. T., Kasibhatla, P., and Malley, C. S.: Magnitude, trends, and impacts of ambient long-term ozone exposure in the United States from 2000 to 2015, Atmos. Chem. Phys., 20, 1757–1775, https://doi.org/10.5194/acp-20-1757-2020, 2020."

**3   Answer to Anonymous Referee #2**

"The authors present a data-driven forecast model for maximum daily 8-hour ozone (mda8O3) concentrations based on multiple convolutional neural layers and apply it to the network of rural ozone monitoring sites in Germany. The manuscript is well

written, and the model presented and its application make a valuable contribution to the field. I ask the authors for a few clarifications and specifications detailed in my specific comments below."

"Specific comments:"

1. "The authors include several meteorological variables as input but not radiation. This is a little odd given the importance of radiation for photochemistry and thus ozone formation. Cloud-cover is used by the authors together with temperature as a proxy,however I am not surprised that this surrogate variable shows limited influence on mda8O3. It seems also that direct and diffuse irradiance are available at varioustime steps (https://reanalysis.meteo.uni-bonn.de/?Download_Data___COSMO-REA6) although otherwise stated on page p4, L93."
   The COSMO-REA6 indeed contains radiative variables, however those variables are unfortunately not available through the JOIN-REST-API on https://join.fz-juelich.de which we used to access the TOAR-database for this study. Re-processing of the additional variables would have been a major effort, which could not be done during the time this study was conducted. We therefore used the mentioned variables as proxy.

2. "Related to my previous comment I am a bit puzzled that all meteorological covariates show such limited added skill. Have the authors assessed simple brute-force correlations between meteorological covariates and mda8O3, are they as low as the skill score would suggest? Also I wonder if the limited influence stems from the joint consideration of all seasons and if a cleaner picture would emerge on seasonal or monthly basis."
   We did not calculate correlations between meteorological covariates and ozone

concentrations, but also Otero et al. (2016) report, that the ozone concentration itself, followed by temperature are the main drivers in their correlation analysis. In general, instead of using the bootstrapped variables as inputs for the trained model, it would also be possible to use those bootstrapped inputs to train the model for each redrawn variable from scratch, resulting in nine different models (one per variable). In this case, the network would be forced to extract more information from variables which are not redrawn. This method, however, would deliver information on the influence of variables during different training cycles and not the influence of variables on the originally trained model. Our method, however, underestimates the influence of specific input variables in case of strongly correlated input variables as the majority of information is extracted by the most dominant feature (here ozone). As ozone dominates Manuscript-Figure 8, we split the variables into two groups and show their skill scores on different scales.

To make this point even more explicit, we added the following half-sentence to page 12, line 330: ", because in such cases the network will focus on the dominant feature (here: ozone)."

3. "Along these lines, I am wondering if pooling of observations in the samples might cause some spurious effects. Given the relatively low VOC abundances during fall-early spring pooling might explain to some degree the relative low skill obtained for NO and NO2."
We use all seasons to train one specific model which should be applicable to any (German) collection of time-series containing the required nine input variables. This generalisation comes with the cost that the network has to learn the seasonality first, before focusing on variational patterns.

4. "On p.5., L103 the authors state that they include stations if they have at least one year of valid data in one of the sets. I wonder if an unequal inclusion of observations from different time periods affects the robustness of the training and tuning. How much would the sample reduce if a more stringent criterion would

be applied say e.g., more than 80% or 50% coverage over the considered time period, or a high fraction of available data per month, season and year?"

We only include a station into a corresponding data set, if at least one year of valid data is available in the set's time period. We expanded Manuscript-Figure 2 and show the total amount of samples available for each day. As additional information to the reviewer, we here also provide Fig. 1 (here) showing the number of stations (y-axis) as a function of valid data points (x-axis).

Furthermore, we extended Manuscript-Table 4, which now also reports the mean and standard deviation as well as selected percentiles. On average each station in the validation set contains $\sim 690$ samples (std. $\sim 60$ samples), while each station in the testing set contains on average $\sim 1040$ samples (std. $\sim 90$ samples). Thus, a more stringent criterion would not lead to a significant reduction of stations in the validation and testing sets. Within the training data set roughly two thirds ($\sim 200$) of all stations contain at least 1500 samples and approximately 120 stations have more than 2500 samples.

5. "Also O3 and NO and NO2 show substantial changes and trends in Europe over the time period considered. The authors do not address this in their manuscript, thus I assume the data has not been detrended before use in model training and validation? What magnitude of effect would we expect by considering non-stationary time series training of a CNN model?"

Indeed, we did not detrend any time-series. We added the following paragraph as a separate section:

By splitting the data into three consecutive, non-overlapping sets, we ensure that the data sets are as independent as possible. On the other hand, this independence comes at the cost, that changes of trends in the input variables may not be captured, especially as our input data are not de-trended. Indeed, at European non-urban measurement sites, several ozone metrics related to high concentrations (e.g. 4th highest daily maximum 8-hour (4MDA8) or the

95%-percentile of hourly concentrations) show a significant decrease during our study period (1997 to 2015) (Fleming et al., 2018; Yan et al., 2018). Our data splitting method for evaluating the generalisation capability is conservative in the sense that we evaluate the model on the test set, which has the largest possible distance to the training set. If our research model shall be transformed into an operational system we suggest to apply online learning and use the latest available data for subsequent training cycles (see for example Sayeed et al. (2020)).
(Same answer to rev #1)

We expect that the change of trend is more important than the trend itself as the change of trend might not be captured in the different sets. As the ozone metrics related to high concentrations are decreasing, the network does not need to predict values outside the learned/trained concentration range in our specific application.

6. "The authors use dma8 for NO and NO2. What is the motivation to use here dma8 instead of the daily mean or maximum value, which would be the more common quantities?"
   While we agree with the reviewer that the choice of metrics for NO and NO2 may seem strange, this was indeed done on purpose. We now added the following text to explain our rationale for this decision:
   "The choice of using the dma8eu metrics for NO and NO2 was motivated by the idea to sample all chemical quantities during the same time periods and with similar averaging times. While the dma8eu metrics is calculated based on data starting at 5pm on the previous day, the daily mean/max values for example would be calculated based on data starting from the current day. To test the impact of using different metrics for ozone precursors we also trained the model from scratch with either mean or maximum concentrations of NO and NO2 as inputs.

The results of these runs were hardly distinguishable from the results presented below."
(Same answer to rev #1)

7. P6, L135, a batch size of 512 samples is used, how do the authors derive this number?
We tested various batch sizes and found no significant differences in the results. We now added the following explanation in the text on page 6, line 134: "[We selected a batch size of 512 samples (Algorithm 1, line 10)], because this size is a good compromise between minimising the loss-function and optimising computing time per trained epoch. Experiments with larger and smaller batch sizes did not yield significantly different results."

"Minor comments:"

1. "I suggest to use 'training set' instead of 'train set' throughout the manuscript."
Thank you for the suggestion, we changed all 'train set' to 'training set'

2. "I suggest grouping several figures to multi-panel figures to increase accessibility (7a-d, A3a-d, A4a-d,A5a-d, A6a-d)."
Done.

3. "Figure A2 is incredibly hard to read even when zooming in to 400%"
We replaced this figure (now A2 and A3) and used a different tool for visualisation of the neural network architecture. We removed manuscript Fig. 3, as the new representation of Figure A2 transports the same content. We adjusted the references on page 6 line 144.

4. "Axis labels of Figure A2 are hard to read."
We assume you mean Figure A1; we increased the font size.

5. "P7, L174 a reference for the Adam optimizer is missing"
   Reference is Kingma and Ba (2014)

6. "I was wondering if the section on joint distributions and skill scores could not be moved to the Appendix"
   We moved the section on joint distributios to the appendix. We would like to keep the section on skill scores in the main text as two figures show that skill score on the y-axis (Manuscript Fig. 5, Fig 6).

"Spelling and typos":

- "P6, L118: hyperparameters"
  We replaced 'hyperparameetrs' with 'hyperparameters'

- "P9L226: replace 'model with 'models'"
  We replaced 'model' with 'models'

- "P10, L235: replace 'observation' with 'observations'"
  We replaced 'observation' with 'observations'

- "P10, L237: replace 'multi-valued' with 'multi-value' and check throughout the text"
  We replaced 'multi-valued' with 'multi-value' throughout the manuscript

- "P11, L295: therefore more credible"
  We added 'more'

- "P11, 297: replace 'the network under-forecasts' with 'the forecast is underestimating'"
  We replaced 'the network under-forecasts' with 'the forecast is underestimating'

[Figure]

**4  Additional Changes**

1. We added the unit to Manuscript-Figure 4

2. We added the following sentence on page 3 line 57: "We evaluate the performance on 204 stations, which have data during the 2010 to 2015 period by looking at skill scores, the joint distribution of forecasts and observations, as well as the influence of input variables."

3. The correct number of stations used in the training set is 312, we corrected this in the manuscript.

4. We updated the Acknowledgements

**5  References**

- Fleming, Z. L., Doherty, R. M., Von Schneidemesser, E., Malley, C. S., Cooper, O. R., Pinto, J. P., Colette, A., Xu, X., Simpson, D.,Schultz, M. G., Lefohn, A. S., Hamad, S., Moolla, R., Solberg, S., and Feng, Z.: Tropospheric Ozone Assessment Report: Present-day ozone distribution and trends relevant to human health, Elem Sci Anth, 6, 12,
https://doi.org/10.1525/elementa.273,
https://www.elementascience.org/article/10.1525/elementa.273/, 2018

- Kingma, D. P. and Ba, J.: Adam: A Method for Stochastic Optimization, arXiv:1412.6980 [cs], http://arxiv.org/abs/1412.6980, arXiv:1412.6980, 2014.

- Otero, N., Sillmann, J., Schnell, J. L., Rust, H. W., and Butler, T.: Synoptic and meteorological drivers of extreme ozone concentrations overEurope,
Environmental Research Letters, 11, 024 005, https://doi.org/10.1088/1748-9326/11/2/024005,
https://iopscience.iop.org/article/10.1088/1748-9326/11/2/024005, 2016.

- Sayeed, A., Choi, Y., Eslami, E., Lops, Y., Roy, A., and Jung, J.: Using a deep convolutional neural network to predict 2017 ozone concen-trations, 24 hours in advance, Neural Networks, 121, 396–408, https://doi.org/10.1016/j.neunet.2019.09.033, https://linkinghub.elsevier.com/retrieve/pii/S0893608019303156, 2020.

- Schultz, M. G., Betancourt, C., Gong, B., Kleinert, F., Langguth, M., Leufen, L. H., Mozaffari, A., and Stadler, S.: Can deep learning beatnumerical weather prediction?, Philosophical Transactions A, accepted, 2020.

- Seltzer, K. M., Shindell, D. T., Kasibhatla, P., and Malley, C. S.: Magnitude, trends, and impacts of ambient long-term ozone exposure inthe United States from 2000 to 2015, Atmospheric Chemistry and Physics, 20, 1757–1775, https://doi.org/10.5194/acp-20-1757-2020, https://acp.copernicus.org/articles/20/1757/2020/, 2020.

- Yan, Y., Pozzer, A., Ojha, N., Lin, J., and Lelieveld, J.: Analysis of European ozone trends in the period 1995–2014, Atmospheric Chemistryand Physics, 18, 5589–5605, https://doi.org/10.5194/acp-18-5589-2018, https://acp.copernicus.org/articles/18/5589/2018/, 2018.

[Figure]

**Fig. 1.** Number of stations having at least a given number of samples.